# ResQ: A Residual Q Function-based Approach for Multi-Agent Reinforcement Learning Value Factorization

**Siqi Shen**[†]**, Mengwei Qiu**[†]**, Jun Liu**[†]**, Weiquan Liu**[†]**, Yongquan Fu**[‡]*****, **Xinwang Liu**[‡]**, Cheng Wang**[†]

[†]Fujian Key Lab of Sensing and Computing for Smart Cities, School of Informatics, Xiamen University, China
[‡]School of Computer, National University of Defense Technology, China
{siqishen,wqliu,cwang}@xmu.edu.cn, {yongquanf,xinwangliu}@nudt.edu.cn
{mengweiqiu,junliu}@stu.xmu.edu.cn

## Abstract

The factorization of state-action value functions for Multi-Agent Reinforcement Learning (MARL) is important. Existing studies are limited by their representation capability, sample efficiency, and approximation error. To address these challenges, we propose, ResQ, a MARL value function factorization method, which can find the optimal joint policy for any state-action value function through residual functions. ResQ masks some state-action value pairs from a joint state-action value function, which is transformed as the sum of a main function and a residual function. ResQ can be used with mean-value and stochastic-value RL. We theoretically show that ResQ can satisfy both the individual global max (IGM) and the distributional IGM principle without representation limitations. Through experiments on matrix games, the predator-prey, and StarCraft benchmarks, we show that ResQ can obtain better results than multiple expected/stochastic value factorization methods.

## 1 Introduction

Many real-world tasks involve multiple agents acting together, such as robot control [1]. Such tasks can be modelled as Multi-Agent Reinforcement Learning (MARL) problems where a group of agents must cooperate to achieve a common goal. MARL has attracted great research interest because of its social and economic impact. However, MARL is highly stochastic and hard to learn due to the partial-observability [2] and changing policies of agents. To address these issues, many approaches adopt the Centralized Training with Decentralized Execution paradigm (CTDE) [3].

In the CTDE paradigm, value factorization approaches [4] are widely adopted due to their remarkable performance and sample efficiency. A joint state-action value function is factorized into per-agent utilities $Q_i$, and each agent $i$ acts greedily according to $Q_i$. To enable effective CTDE, it is critical to ensure the Individual-Global-Max (IGM) principle [5] that the optimal joint action should be equivalent to the collection of each agent's greedy actions. VDN [6] and QMIX [4] are two popular value factorization methods, which satisfy the IGM principle.

Most value factorization approaches focus on the deterministic state-action value functions. Albeit MARL is highly stochastic, the full distribution of per-agent utilities and the multi-agent system are overlooked and represented only as expected value of the full distribution. Such distributional information could be beneficial for policy learning. DMIX and DDN [7] extend QMIX and VDN with distributional RL. They can factorize stochastic joint state-action value pairs into stochastic per-agent utilities, and satisfy the distributional individual-global-max(DIGM) principle [7]. However, because

---

*Corresponding author

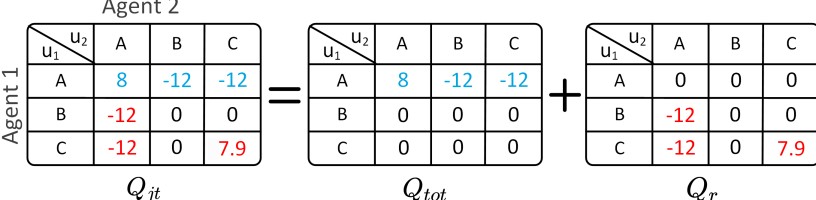

Figure 1: Motivating Example of ResQ. The reward (state-action value function) of a one-step game, denoted as $Q_{jt}(u_1, u_2)$. The game consists of two agents; each has three actions (A, B, and C). The optimal policy is to choose action $A$ for both two agents. This is a non-monotonic matrix. If agent 1 chooses action A, the reward vector for agent 2 becomes [8, -12, -12]. It monotonically increases from the right to the left. If agent 1 chooses action B/C, the rewards ([-12, 0, 0] and [-12, 0, 7.9]) for agent 2 monotonically increase from the left to the right. The direction of increment is different between the first and the second/third action.

of the representation limitations of QMIX and VDN, DMIX and DDN cannot factorize distributional value functions whose mean value is non-monotonic.

To address the representation limitations, QTRAN [5], WQMIX [8], and QPLEX [9] have been proposed. However, QTRAN and QPLEX are sample-inefficient, and WQMIX has high approximation errors to non-optimal actions. These methods may lead to incorrect estimations of optimal values. Moreover, they satisfy only the IGM principle. Achieving the IGM and the DIGM principles without representation limitations remains an open challenge.

We address these challenges through, *ResQ*, a residual Q function-based approach. Given a hard-to-factorize joint state-action value function $Q_{jt}$ (e.g., shown in Fig. 1), the key insight of our work is described as follows. We could obtain an easy-to-factorize state-action value function, the main function $Q_{tot}$, if we can mask out some state-action value pairs from $Q_{jt}$. If the main function $Q_{tot}$ shares the same optimal policy with $Q_{jt}$, then we can obtain the optimal policy of $Q_{jt}$ through the factorization of $Q_{tot}$ by just using some simple mixers (e.g., [6]). After the masking, $Q_{jt}$ is divided into two value functions: a main function $Q_{tot}$ and a residual function $Q_r$. $Q_r$ are the masked state-action value pairs from $Q_{jt}$.

Besides mean-value RL (e.g., $Q_{jt}$), the idea of ResQ can be used with distributional RL [10] as well. ResQ can factorize stochastic joint state-action value functions $Z_{jt}$ into per-agent stochastic utilities $Z_i$ with a mask and a residual function $Z_r$. We have shown that ResQ satisfies both *the IGM and the DIGM principles* without representation limitations.

For evaluation, we conduct extensive experiments on one-step matrix games, the StarCraft II MARL tasks [11], and predator-prey tasks. The experimental results show that ResQ can obtain better results than multiple competitive value factorization methods, and the ablation study shows that through the use of residual functions for multiple factorization methods, ResQ can improve the performance of these methods. The source code of ResQ can be visited at https://github.com/xmu-rl-3dv/ResQ.

## 2 Background

### 2.1 Dec-POMDPs

We consider cooperative Multi-Agent Reinforcement Learning (MARL) scenarios which can be modelled as Decentralized Partially Observable Markov Decision Processes (Dec-POMDPs) [12] defined as $G$ for $n$ agents. $G = \langle \mathcal{S}, \{\mathcal{U}_i\}_{i=1}^n, P, r, \{\mathcal{O}_i\}_{i=1}^n, \{\sigma_i\}_{i=1}^n, n, \gamma \rangle$. $\mathcal{S}$ denotes the set of states and $\mathcal{U}_i$ the set of actions available to agent $i$ and we consider discrete actions only. A joint action of all agents is defined as $\boldsymbol{u}^t \in \mathcal{U}^N := \mathcal{U}_1 \times \ldots \times \mathcal{U}_n$. At a discrete time step $t$ and state $s^t$, after the joint action is issued, the next state $s^{t+1} \in \mathcal{S}$ of the environment is drawn from the transition function $s^{t+1} \sim P(\cdot|s^t, \boldsymbol{u}^t)$. All the agents receive a reward $r^t$ after the state transition happens. It is a partially observable environment that agent $i$ observes a part of the environment $o_i^t \in \mathcal{O}_i$ which is drawn from $o_i^t \sim \sigma^i(\cdot|s^t)$. $\gamma$ is the discount factor. Each agent $i$ maintains an

action-observation history $\tau_i = (O_i \times U_i)^{*1}$, and acts according to policy $\pi_i(u_i|\tau_i)$. We denote $\tau \in \mathcal{T}^N := \tau_1 \times \ldots \times \tau_n$ as the joint-observation history. The learning objective is to find the optimal policy $\pi = <\pi_1, ..., \pi_n>$.

## 2.2 Value Function Factorization

The Individual-Global-Max (**IGM**) principle proposed in [5] is important to realize the factorization of MARL state-action value function. It is defined as follows.

**Definition 1** (IGM). *For a joint state-action value function $Q_{\mathrm{jt}} : \mathcal{T}^N \times \mathcal{U}^N \mapsto \mathbb{R}$, where $\tau \in \mathcal{T}^N$ is a joint action-observation history, if there exist individual state-action functions $[Q_i : \mathcal{T}_i \times \mathcal{U}_i \mapsto \mathbb{R}]_{i=1}^N$, such that the following conditions are satisfied*

$$\arg\max_{\mathbf{u}} Q_{\mathrm{jt}}(\boldsymbol{\tau}, \mathbf{u}) = (\arg\max_{u_1} Q_1(\tau_1, u_1), \ \ldots, \ \arg\max_{u_n} Q_n(\tau_n, u_n)), \tag{1}$$

*then, $[Q_i]$ satisfy IGM for $Q_{\mathrm{jt}}$ under $\tau$. We can state that $Q_{\mathrm{jt}}(\boldsymbol{\tau}, \mathbf{u})$ is factorized by $[Q_i(\tau_i, u_i)]_{i=1}^N$. In this work, we assume the argmax operator is unique, the action with smallest index is selected to break ties if a tie exists*

QMIX is a widely used monotonically increasing mixer function which proposes the following sufficient conditions for IGM:

$$(\text{Monotonicity}) \quad \partial Q_{jt}(\boldsymbol{\tau}, \boldsymbol{u})/\partial Q_i(\tau_i, u_i) \geq 0, \quad \forall i \in \mathcal{N}. \tag{2}$$

There are tasks whose joint state-action value function cannot be expressed well by monotonic increasing conditions, as shown in Fig. 1.

## 2.3 Distributional RL

Distributional RL models the stochastic return of state-action pair through $Z(\boldsymbol{\tau}, \boldsymbol{u})$ explicitly. They models full return distribution $Z(\boldsymbol{\tau}, \boldsymbol{u})$ instead of $Q(\boldsymbol{\tau}, \boldsymbol{u})$. The distribution of return can be approximated through a categorical distribution [13] or a quantile function [10]. Implicit Quantile Function (IQN) [10] models the stochastic value function $Z(\tau, u)$ as a quantile function $F^{-1}(\tau, u|w)$, where $F^{-1}$ is the generalized inverse cumulative distribution function (CDF), $w \in [0, 1]$ is a quantile sample. IQN defined a distributional Bellman operator, and use it to updates its $Z(\boldsymbol{\tau}, \boldsymbol{u})$. After applying the distributional Bellman operator on $Z(\boldsymbol{\tau}, \boldsymbol{u})$, its resulting $Z(\boldsymbol{\tau}', \boldsymbol{u}')$ remains in the same distribution as $Z(\boldsymbol{\tau}, \boldsymbol{u})$. The loss $\rho_t^{w_i, w_j}$ for temporal-difference error $\delta_t^{w_i, w_j}$ is defined as Huber quantile regression loss, where $w_i$ and $w_j$ are two quantiles. During execution, the action with the largest expected return $\arg\max_u \mathbb{E}[Z(\boldsymbol{\tau}, \boldsymbol{u})]$ is chosen.

MARL is highly stochastic, and distributional RL could be used to deal with the stochasticity of MARL. [7] proposes the Distributional Individual-Global-Max (**DIGM**) principle, which is defined as follows.

**Definition 2** (DIGM). *Given a set of stochastic individual state-action value function $[Z_i(\tau_i, u_i)]_{i=1}^N$ and a stochastic joint state-action value function $Z_{jt}(\boldsymbol{\tau}, \boldsymbol{u})$, if the following conditions are satisfied*

$$\arg\max_{\mathbf{u}} \mathbb{E}[Z_{jt}(\boldsymbol{\tau}, \boldsymbol{u})] = (\arg\max_{u_1} \mathbb{E}[Z_1(\tau_1, u_1)], \ \ldots, \ \arg\max_{u_n} \mathbb{E}[Z_n(\tau_n, u_n)]), \tag{3}$$

*then, $[Z_i(\tau_i, u_i)]_{i=1}^N$ satisfy DIGM for $Z_{jt}$ under $\boldsymbol{\tau}$. We can state that $Z_{jt}(\boldsymbol{\tau}, \boldsymbol{u})$ is distributionally factorized by $[Z_i(\tau_i, u_i)]_{i=1}^N$.*

# 3 Related Work

Value factorization approaches are widely adopted in MARL [14]. VDN [6] factorizes the value function $Q_{tot}$ as the sum of per-agents' utility $Q_i$. QMIX [4] supports monotonic relationships among $Q_i$ and $Q_{tot}$. During execution, each agent acts greedily according to $Q_i$. RMIX [15] learns return distribution of each agent, and integrates risk-sensitive RL with QMIX. [7] proposes DMIX and

---

[1]* represents 0 to T, where T denotes the time step. * means that the history could be short or long. For example, $\tau_1$ could $\in O_1^1 \times U_1^1 \times O_1^2 \times U_1^2$ and could $\in O_1^1 \times U_1^1$, where the superscript represent time, and the subscript is the index of a agent.

DDN, which extend QMIX and VDN with distributional RL through mean-shape decomposition. They satisfy the DIGM principle but suffer from the representation limitation of QMIX and VDN. They cannot model stochastic value functions whose mean-value are non-monotonic. QAtten [16] and REFIL [17] adopt the attention mechanisms to focus on certain agents/scenarios when factorizing value function. QRelation [18] considers the relationships among agents for value factorization. However, they cannot represent non-monotonic value functions well.

To address the representational limitations, WQMIX [8] prioritizes the estimation of the optimal state-action value. It assigns higher weights to the optimal joint actions when minimizing approximation errors. WQMIX can be viewed as a special version of ResQ. It pays attention to the optimal actions via masking out all the sub-optimal actions. The $Q_r$ of WQMIX consists of all the sub-optimal state-action pairs, and the $Q_{tot}$ models only the optimal ones. However, it assigns high learning priorities to $Q_{tot}$, which puts the learning of sub-optimal state-action pairs and $Q_{jt}$ in trouble. Moreover, it is difficult to assign proper weights to obtain satisfactory performance.

QTRAN [5] and QTRAN++ [19] transform the joint state-action value function $Q_{jt}$ into an easy-to-factorize one through a series of linear constraints. QTRAN learns an approximated function $Q_{tran}(\boldsymbol{\tau}, \boldsymbol{u}) = \sum_{i=1}^{N} Q_i(\tau_i, u_i) + V_{jt}(\boldsymbol{\tau})$ to approximate the optimal policy of $Q_{jt}$. They can be viewed as using $Q_{tot} = \sum_{i=1}^{N} Q_i(\tau_i, u_i)$ to model the optimal actions, and constraining $Q_{tran} \geq Q_{jt}$ for sub-optimal actions. QTRAN use soft constraints (MSE losses) to implement IGM which could lead to the violation of IGM.

RQN [20] extends QTRAN by adding an individual correction factor for each utility function to compute an adjusted utility function. RQN can be viewed as a special case of ResQ. It uses the sum of per-agent utility as the main function $Q_{tot}$, and the sum of individual correction factors as the residual function $Q_r$.

QPlex decomposes $Q_{jt}$ as the sum of a value function $V_{tot}$ and a non-positive advantage function $A_{tot}$. The role of $A_{tot}$ is similar to the main function $Q_{tot}$ used in ResQ. QPlex obtains agents' policies from the factorization of $A_{tot}$, which must be $\leq 0$, but ResQ does not have such restrictions. Although QPlex does not suffer from representation limitation issues, it may struggle to learn a good policy in complex tasks [8].

MAVEN [21] deals with the inefficient exploration problem in MARL. Inefficient exploration problems could interact with the representation limitation problems. It adopts QMIX as its value factorization method. CDS [22] improves the effective cooperation of agents with efficient exploration. Different from them, ResQ focus on value factorization, and can be used in MAVEN and CDS.

COMNet [23], DIAL [24], and GraphComm [25] study communication among agents. DCG [26] uses graphs for coordination. COPA [27] and NDQ [28] factorize the value function with communication. The communication method adopted by them could be used with ResQ to improve MARL performance. There exist various MARL actor-critic methods, such as MADDPG [29], MAAC [30], and COMA [31]. ResQ is a value-based method. Other methods exist. For example, MAPPO [32] combines MARL with PPO [33]. UPDeT [34] proposes a MARL policy decoupling method. ResQ is orthogonal to them.

## 4 Method

**Motivating Example** Fig. 1 shows an example of a non-monotonic payoff matrix, which is hard-to-factorized. Agent 1 has different action-value orderings, which depend on the action of Agent 2. Moreover, the second-best state-action value 7.9 is only slightly lower than the optimal value 8, which causes difficulties for methods with representation limitations or overestimations.

Given a hard-to-factorize state-action value function $Q_{jt}$, the key insight to ResQ is that if we can mask out some state-action value pairs from $Q_{jt}$, then we could obtain an easy-to-factorize value function $Q_{tot}$. As shown in Fig. 1, if we mask out those action value pairs colored in red, then the main function $Q_{tot}$ can be factorized easily by mixers such as QAtten or QMIX. ResQ seeks to find a mask $w$, a main functions $Q_{tot}$, and a residual function $Q_r$ which satisfy the following equation.

$$Q_{jt}(\boldsymbol{\tau}, \boldsymbol{u}) = w_{tot}(\boldsymbol{\tau}, \boldsymbol{u}) Q_{tot}(\boldsymbol{\tau}, \boldsymbol{u}) + w_r(\boldsymbol{\tau}, \boldsymbol{u}) Q_r(\boldsymbol{\tau}, \boldsymbol{u}) \tag{4}$$

where masks $w_{tot}(\boldsymbol{\tau}, \boldsymbol{u})$, $w_r(\boldsymbol{\tau}, \boldsymbol{u}) \in \{0, 1\}$. The main $Q_{tot}$ shares the same greedy optimal policy as $Q_{jt}$. $Q_{tot}(\boldsymbol{\tau}, \boldsymbol{u})$ is factorized into per-agent utilities $Q_i$, and each agent selects its action greedily

with respect to $Q_i$. There are many possibilities to construct the masks $w_{tot}$ and $w_r$, in this work, we set $w_{tot}$ to be constant 1, and focus on $w_r$.

We describe how ResQ factorizes an expected-value joint state-action value function $Q_{jt}$ into per-agent utilities in Sec. 4.1, and how ResQ decomposes a stochastic joint state-action value function $Z_{jt}$ into stochastic individual utilities in Sec. 4.2.

## 4.1 Residual Q

For a given joint action-observation history $\boldsymbol{\tau}$, consider a factorizable joint state-action value function $Q_{jt}(\boldsymbol{\tau}, \boldsymbol{u})$ that can be expressed in the form as (5). Let $\bar{u}_i = \arg\max_{u_i} Q_i(\tau_i, u_i)$, $\bar{\boldsymbol{u}} = [\bar{u}_i]_{i=1}^N$, Theorem 1 states the sufficient conditions for state-action value functions $[Q_i(\tau_i, u_i)]_{i=1}^N$ that satisfy the IGM principle for $Q_{jt}$.

**Theorem 1.** *A joint state-action function*

$$Q_{jt}(\boldsymbol{\tau}, \boldsymbol{u}) = Q_{tot}(\boldsymbol{\tau}, \boldsymbol{u}) + w_r(\boldsymbol{\tau}, \boldsymbol{u})Q_r(\boldsymbol{\tau}, \boldsymbol{u}) \tag{5}$$

*is factorized by* $[Q_i(\tau_i, u_i)]_{i=1}^N$, *if* $Q_r(\boldsymbol{\tau}, \boldsymbol{u}) \leq 0$, $Q_{tot}(\boldsymbol{\tau}, \boldsymbol{u})$ *and* $[Q_i(\tau_i, u_i)]_{i=1}^N$ *satisfy the monotonicity conditions (2), and*

$$w_r(\boldsymbol{\tau}, \boldsymbol{u}) = \begin{cases} 0 & \boldsymbol{u} = \bar{\boldsymbol{u}}, & \text{(6a)} \\ 1 & \boldsymbol{u} \neq \bar{\boldsymbol{u}}, & \text{(6b)} \end{cases}$$

The proof of Theorem 1 is shown in the appendix. In Theorem 1, we have shown that for any factorizable value function $Q_{jt}$ that can be expressed as (5), we can find a main function $Q_{tot}$ that shares the same optimal policy as $Q_{jt}$. In Theorem 2, we show that for any joint state-action value function $Q$, we can find a joint state-action value function $Q_{jt}$ that can be expressed as (5), and it has the same optimal policy as $Q$.

**Theorem 2.** *For any joint state-action function* $Q(\boldsymbol{\tau}, \boldsymbol{u})$, *we can find* $Q_{jt}(\boldsymbol{\tau}, \boldsymbol{u}) = Q_{tot}(\boldsymbol{\tau}, \boldsymbol{u}) + w_r(\boldsymbol{\tau}, \boldsymbol{u})Q_r(\boldsymbol{\tau}, \boldsymbol{u})$ *that*

$$\bar{u} = \arg\max_u Q(\boldsymbol{\tau}, \boldsymbol{u}) = \arg\max_u Q_{jt}(\boldsymbol{\tau}, \boldsymbol{u}) \tag{7}$$

$$Q(\boldsymbol{\tau}, \boldsymbol{u}) = Q_{jt}(\boldsymbol{\tau}, \boldsymbol{u}) \quad \forall \boldsymbol{u} \neq \bar{\boldsymbol{u}} \tag{8}$$

$Q_{tot}(\boldsymbol{\tau}, \boldsymbol{u})$ *monotonically increases with* $[Q_i(\tau_i, u_i)]_{i=1}^N$, $w_r(\boldsymbol{\tau}, \boldsymbol{u})$ *satisfies (6), and* $Q_r(\boldsymbol{\tau}, \boldsymbol{u}) \leq 0$.

The proof of Theorem 2 and other theorems are provided in the appendix. From Theorem 2, for any joint state-action function $Q(\boldsymbol{\tau}, \boldsymbol{u})$, we can find $Q_{jt}(\boldsymbol{\tau}, \boldsymbol{u})$ that shares the same optimal policy as $Q$. Moreover, $Q_{jt}$ can be expressed as (5). Combining Theorem 1 and 2, for any $Q(\boldsymbol{\tau}, \boldsymbol{u})$, we can find $[Q_i(\tau_i, u_i)]_{i=1}^N$ satisfy the IGM principle for $Q(\boldsymbol{\tau}, \boldsymbol{u})$ through a residual function $Q_r(\boldsymbol{\tau}, \boldsymbol{u}) \leq 0$, a main function $Q_{tot}(\boldsymbol{\tau}, \boldsymbol{u})$, and a mask function $w_r(\boldsymbol{\tau}, \boldsymbol{u}) \in \{0, 1\}$. The policies $[Q_i(\tau_i, u_i)]_{i=1}^N$ can be obtained through factorizing $Q_{tot}(\boldsymbol{\tau}, \boldsymbol{u})$ using mixers such as QMIX [4]. Furthermore, Formula (8) in Theorem 2 shows that $Q_{jt}(\boldsymbol{\tau}, \boldsymbol{u})$ can track $Q(\boldsymbol{\tau}, \boldsymbol{u})$ closely $\forall \boldsymbol{u} \neq \bar{\boldsymbol{u}}$. Thus, ResQ can model Q values precisely.

## 4.2 Distributional Residual Q

One promising way to deal with the MARL stochasticity is to model the return of state-action pairs as distributional values [10]. Researchers [7] have proposed a distributional MARL approach that satisfies the DIGM principle. However, it cannot model stochastic joint value functions whose value-expectation are non-monotonic. ResQ can address this issue through using $Z_{dmix}(\boldsymbol{\tau}, \boldsymbol{u})$ as the main function and $Z_r$ as the residual function, where $Z_{dmix}(\boldsymbol{\tau}, \boldsymbol{u})$ is the factorization function of DMIX or DDN [7], and $Z_{dmix}, Z_r$ are both stochastic value functions. Formula (9) in Theorem **??** describes how $Z_{jt}$ is decomposed.

**Theorem 3.** *A stochastic joint state-action function*

$$Z_{jt}(\boldsymbol{\tau}, \boldsymbol{u}) = Z_{dmix}(\boldsymbol{\tau}, \boldsymbol{u}) + w_r(\boldsymbol{\tau}, \boldsymbol{u})Z_r(\boldsymbol{\tau}, \boldsymbol{u}) \tag{9}$$

*is factorized by* $[Z_i(\tau_i, u_i)]_{i=1}^N$, *if* $Z_r(\boldsymbol{\tau}, \boldsymbol{u}) \leq 0$ *and* $w_r(\boldsymbol{\tau}, \boldsymbol{u}) = 0$ *when* $\boldsymbol{u} = \bar{\boldsymbol{u}}$, *otherwise* 1. $\bar{u}_i = \arg\max_{u_i} \mathbb{E}[Z_i(\tau_i, u_i)]$, $\bar{u} = [\bar{u}_i]_{i=1}^N$, $Z_{dmix}(\boldsymbol{\tau}, \boldsymbol{u}) = Z_{mean}(\boldsymbol{\tau}, \boldsymbol{u}) + Z_{shape}(\boldsymbol{\tau}, \boldsymbol{u})$, $\mathbb{E}[Z_{shape}(\boldsymbol{\tau}, \boldsymbol{u})] = 0$, $Q_i = \mathbb{E}[Z_i(\tau_i, u_i)]$. $Z_{mean}(\boldsymbol{\tau}, \boldsymbol{u})$ *is a monotonic increasing function with respect to* $Q_i$.

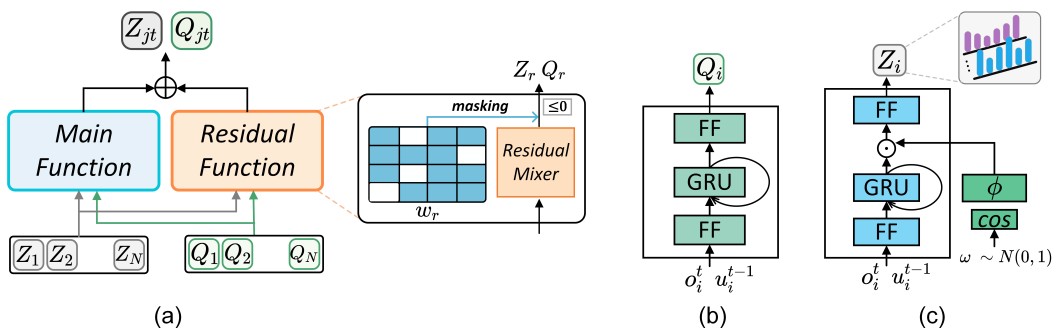

Figure 2: The architecture of ResQ. (a): the main and the residual function, (b): agent utility, (c): stochastic agent utility

where $Z_{mean}(\boldsymbol{\tau}, \boldsymbol{u})$ models the mean of $Z_{dmix}$. In DMIX [7], it is implemented using QMIX, and in DDN, it is implemented as VDN. $Z_{shape}$ models the variation of $Z_{dmix}$. Theorem 3 shows that ResQ can remove the representation limitations of DMIX.

Besides freeing [7] from representation limitations, ResQ can factorize a stochastic value function $Z_{jt}$ into individual stochastic utilities $Z_i$ with a linear main function $Z_{tot}$, a residual function $Z_r$, and a mask function $w_r$. Theorem 4 states the sufficient condition for $[Z_i(\tau_i, u_i)]_{i=1}^N$ satify the DIGM principle.

**Theorem 4.** *A stochastic joint state-action function*

$$Z_{jt}(\boldsymbol{\tau}, \boldsymbol{u}) = Z_{tot}(\boldsymbol{\tau}, \boldsymbol{u}) + w_r(\boldsymbol{\tau}, \boldsymbol{u})Z_r(\boldsymbol{\tau}, \boldsymbol{u}) \tag{10}$$

*is factorized by $[Z_i(\tau_i, u_i)]_{i=1}^N$, if $Z_r(\boldsymbol{\tau}, \boldsymbol{u}) \leq 0$, $Z_{tot}(\boldsymbol{\tau}, \boldsymbol{u}) = \sum_{i=1}^N k_i Z_i(\tau_i, u_i)$ $k_i \geq 0$ and $w_r(\boldsymbol{\tau}, \boldsymbol{u}) = 0$ when $\boldsymbol{u} = \bar{\boldsymbol{u}}$, otherwise 1, where $\bar{\boldsymbol{u}} = [\bar{u}_i]_{i=1}^N$ $\bar{u}_i = \arg\max_{u_i} \mathbb{E}[Z_i(\tau_i, u_i)]$.*

The proofs of Theorem 3 and 4 are provided in the appendix. Theorem 4 states that ResQ can find individual stochastic utilities satisfying the DIGM principle for $Z_{jt}$. And it does not limit the representation capacity of stochastic joint state-action function $Z_{jt}$.

In (10), the main function $Z_{tot}$ is a positive weighted sum of $Z_i$ instead of a monotonic mixing individual utilities as in (5). This is because not all monotonic increasing functions can be used as the main function. Some monotonic increasing function (e.g., $Z_{tot} = \sum_{i=1}^N Z_i^2$) could lead to incorrect estimation of the optimal actions (please refer to the Appendix for an example), they cannot satisfy the DIGM principle.

### 4.3 Neural Networks

For any state-action value function $Q_{jt}$, if we can construct the main function $Q_{tot}$, the mask $w_r$, and the residual function $Q_r$ as (5), then we can recover the optimal policy for $Q_{jt}$. Similarly, for stochastic function $Z_{jt}$, we can recover the optimal actions for $Z_{jt}$ through the main function $Z_{tot}$, the residual function $Z_r$ and the mask as (10). The neural network of ResQ is depicted in Fig. 2.

**Main Function** For $Q_{jt}$, $Q_{tot}(\boldsymbol{\tau}, \boldsymbol{u})$ can be any monotonically increasing function. The most well known monotonic increasing function of per-agent utility $[Q_i]_{i=1}^N$ are QMIX, VDN, and QAtten. In default, ResQ uses QMIX as $Q_{tot}$. For the stochastic case, the stochastic main function $Z_{tot}(\boldsymbol{\tau}, \boldsymbol{u})$ can be a positive weighted sum of individual utilities $Z_i(\tau_i, u_i)$. $Z_{tot}(\boldsymbol{\tau}, \boldsymbol{u}) = \sum_{i=1}^N k_i Z_i(\tau_i, u_i)$ $k_i \geq 0$. Specifically, we model $k_i$ by the attention mechanism [16].

**Residual Function** The residual function $Q_r(\boldsymbol{\tau}, \boldsymbol{u})$ is a feed-forward network that takes action-observation history $\tau$, agents' utilities $Q_i$, and action $[u_i]_{i=1}^N$ as input. Unlike $Q_{tot}$, $Q_r$ is not constrained to be monotonic or positive weighted-sum of utilities, thus allowing it to model a larger class of functions. For $[Q_i]_{i=1}^N$, $Q_r$ is a simple feed-forward network which takes $[Q_i]_{i=1}^N$ as inputs, and the negative absolute function is the activation function of the last neural network layer to ensure $Q_r \leq 0$. For the stochastic case, the residual function $Z_r$ is modeled as $-|\sum_{i=1}^N w_i Z_i|$.

Table 1: Studied Algorithms

| Value type | Algorithms |
|---|---|
| Expected value | ResQ, QMIX [4], CW QMIX [8], OW QMIX [8], REFILL [17], QTran [5], QPlex [9], QAtten [16], VDN [6] |
| Stochastic value | ResZ, DAtten, DMIX [7], DDN [7] |

**Mask Function** The mask $w_r(\boldsymbol{\tau}, \boldsymbol{u})$ requires knowing the maximal joint actions for the joint state-action value function. However, it is intractable to obtain the optimal actions without searching over the entire action space. For practical implementation, we use approximation, and $w_r$ is defined as follows.

$$w_r(\boldsymbol{\tau}, \boldsymbol{u}) = \left\{ \begin{array}{ll} 0 & \boldsymbol{u} = \widetilde{\boldsymbol{u}} \quad (11a) \\ 1 & \boldsymbol{u} \neq \widetilde{\boldsymbol{u}} \quad (11b) \end{array} \right.$$

where $\widetilde{u} = \arg\max_u Q_{tot}(\tau, u)$ for the expectation case. For the stochastic case, $\widetilde{u} = \arg\max_u \mathbb{E}[Z_{tot}(\boldsymbol{\tau}, \boldsymbol{u})]$.

**Loss** For the expected-value case, the loss $L$ of ResQ consists of two temporal-difference losses: $L^*$ and $L^{jt}$. $L = L^* + L^{jt}$. $L^* = \sum_{k=1}^b (y^k - Q_{jt}(\tau^k, u^k, \theta))^2$, where $y^k = r + \gamma Q_{jt}(\tau^{k+1}, \widetilde{u}, \theta^-)$, $\widetilde{u} = [\widetilde{u}_i]_{i=1}^N$, $\widetilde{u}_i = \arg\max Q_i(\tau_i^{k+1}, u_i)$. $\theta^-$ are the parameters of a target network. $L^{jt}$ measure the distance between $Q_{jt}$ and $Q_{tot} + w_r Q_r$. It is defined as $L^{jt} = \sum (Q_{jt} - Q_{tot} - w_r Q_r)^2$. For the distributional case, $L = L^{z*} + L^{zjt}$. $L^{z*}$ and $L^{zjt}$ are similar to $L^*$ and $L^{jt}$, respectively. $L^{z*}$ and $L^{zjt}$ use the pair-wise sampled temporal difference error instead of the standard temporal error, and the Huber quantile regression loss [10] rather than the mean square error loss.

## 5 Evaluation

We evaluate the performance of ResQ on one-step matrix games, the StarCraft II Multi-Agent Challenge benchmark (SMAC) [11], and predator-prey with multiple algorithms. The experimental results show that ResQ can obtain *better results* than state-of-the-art methods, and it satisfies both the *IGM and DIGM principle* together without representation limitation. The ablation study shows that ResQ can *improve the performance* of multiple value factorization methods through using the residual function. Please refer to the appendix for detailed setup and more results.

### 5.1 Experimental Setup

The studied algorithms are listed in Table 1. ResZ is the distributional version of ResQ factorized according to (10). DAtten, designed by us, integrates distributional RL [10] and the attention mechanism. Other algorithms are configured with their default parameters, and the number of rollouts is one. Each experiment is repeated at least 5 times with different seeds. The configuration of ResQ follows WQMIX [8].

### 5.2 Matrix Game

We study the representation power of multiple methods for a matrix game (depicted in Fig. 1 and Table 2). All the algorithms are ran through a full exploration $\epsilon = 1$ for $\epsilon$-greedy conducted over 50,000 steps. This setting guarantees the exploration of all possible actions.

Table 2 depicts the $Q_{jt}$ of multiple methods. As it is observed in the table, only ResQ/ResZ, QTran, and CW QMIX can obtain the optimal policy, DMIX, DDN, and QPlex learn the second-best policy, and OW QMIX learns a wrong policy. Albeit CW QMIX can find the optimal action, its learned $Q_{jt}$ has a high approximation error, as it focuses on the optimal actions only. The detailed factorization results are depictd in Table **??** in the appendix. Further, we modify the matrix into a distributional matrix by adding a nomral distribution (mean = 0, std=1) value into all action-value. The payoff matrix and the reconstructed Q values are shown in Table 3 in the appendix. We find that ResQ/ResZ and QTran can find the optimal policy while most of the algorithms cannot. These results demonstrate the ability of ResQ to factorize difficult state-action value functions with low approximation errors without representation limitations.

Table 2: Payoff matrix of a one-step matrix game and reconstructed value function to approximate the optimal policy. Boldface means greedy actions. Red color indicates *wrongly* estimated optimal actions, whereas blue color represents the opposite.

| $u_2$ \ $u_1$ | **A** | **B** | **C** |
|---|---|---|---|
| **A** | **8** | -12 | -12 |
| B | -12 | 0 | 0 |
| C | -12 | 0 | 7.9 |

(a) Game Payoff matrix.

| $Q_2$ \ $Q_1$ | **0.108 (A)** | -0.300 (B) | 0.106 (C) |
|---|---|---|---|
| **0.108(A)** | 8.03 | -12.00 | -11.99 |
| -0.300(B) | -12.00 | 0.00 | 0.00 |
| 0.106(C) | -12.00 | 0.00 | 7.87 |

(b) ResQ: $Q_1, Q_2, Q_{jt}$

| $Z_2$ \ $Z_1$ | **0.82(A)** | -0.77(B) | 0.77(C) |
|---|---|---|---|
| **0.82(A)** | 7.96 | -12.37 | -12.37 |
| -0.77(B) | -12.13 | -0.27 | -0.38 |
| 0.77(C) | -12.22 | -0.27 | 7.86 |

(c) ResZ: $\mathbb{E}[Z_{tot}], \mathbb{E}[Z_1], \mathbb{E}[Z_2]$

| $Q_2$ \ $Q_1$ | -6.07(A) | -0.07(B) | **0.04(C)** |
|---|---|---|---|
| -6.09(A) | -10.88 | -9.99 | -9.93 |
| -0.07(B) | -9.92 | -0.20 | 0.16 |
| **0.04(C)** | -9.85 | 0.15 | **7.81** |

(d) DMIX: $Q_1, Q_2, Q_{jt}$

| $Q_2$ \ $Q_1$ | -6.70(A) | -0.23(B) | **1.45(C)** |
|---|---|---|---|
| -6.70(A) | -13.40 | -6.94 | -5.25 |
| –0.24(B) | -6.93 | -0.47 | 1.22 |
| **1.45(C)** | -5.25 | 1.22 | **2.91** |

(e) DDN: $Q_1, Q_2, Q_{jt}$

| $Q_2$ \ $Q_1$ | **3.48(A)** | 0.15(B) | 3.46(C) |
|---|---|---|---|
| **3.27(A)** | 8.00 | 4.67 | 7.98 |
| 0.15(B) | 4.88 | 1.55 | 4.86 |
| 3.26(C) | 7.99 | 4.65 | 7.97 |

(f) QTran: $Q_1, Q_2, Q_{jt}$

| $Q_2$ \ $Q_1$ | 0.07(A) | -150(B) | **0.08(C)** |
|---|---|---|---|
| 0.07(A) | 15.7 | -3.72 | 0.34 |
| -150(B) | -2.62 | 12.66 | 12.65 |
| **0.08(C)** | -1.20 | 12.44 | **15.83** |

(g) QPlex: $Q_1, Q_2, Q_{jt}$

| $Q_2$ \ $Q_1$ | **0.17(A)** | -25.72(B) | -25.74(C) |
|---|---|---|---|
| **0.17(A)** | 8.00 | -5.04 | -5.04 |
| -24.55(B) | -5.04 | -5.04 | -5.04 |
| -24.55(C) | -5.04 | -5.04 | -5.04 |

(h) CW QMIX: $Q_1, Q_2, Q_{jt}$

| $Q_2$ \ $Q_1$ | -0.03(A) | -50.79(B) | **0.26(C)** |
|---|---|---|---|
| **0.22(A)** | 6.07 | -0.87 | **6.86** |
| -50.32(B) | -0.86 | -0.87 | -0.16 |
| 0.04(C) | 5.49 | -0.87 | 6.29 |

(i) OW QMIX: $Q_1, Q_2, Q_{jt}$

Figure 3: The Test Win Rate of the SMAC benchmarks. The horizontal axis shows the training time step.

## 5.3 StarCraft II Multi-Agent Challenge (SMAC)

In SMAC [11], two teams of agents fight against each other. For the SMAC tasks, we train each algorithm for 1 to 2 million steps, and for every 10,000 steps, the learned policies are evaluated. The test win rate, which measures the average win ratio of the agents controlled by MARL algorithms for each 32 test episodes, is reported.

As shown in Figure 3 (a), ResQ and ResZ achieve the best performance, and QMIX has the third-best results in the MMM2 scenario. The performance improvement over QMIX comes from the fact that ResQ/ResZ can model better the non-monotonic state-action function of the MMM2 scenario. For the MMM scenario, ResQ/ResZ has the best test win rate.

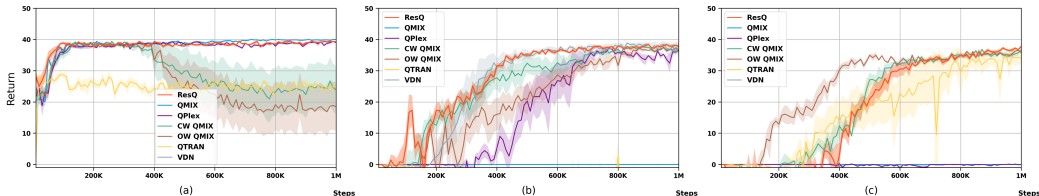

Figure 4: The return of Predator Prey with punishment: $p = 0$ (Left), $p = -2$ (Middle) and $p = -4$ (Right).

The results for the 3s_vs_5z tasks are depicted in Fig. 3 (c). ResQ can obtain close-to-optimal performance after 1 million steps, while others cannot. For the bane_vs_bane scenario, in Fig. 3 (d), ResQ and ResZ can learn the optimal policy in less than 0.2 million steps, while the other methods cannot. The results indicate that through masking state-action value pair from $Q_{jt}/Z_{jt}$, ResQ/ResZ can learn the optimal policy in short time, and the performance is *stable*. Albeit QPlex and DMIX can obtain close-to-optimal performance at a much later time, but their performance is unstable. The results for CW/OW QMIX for bane_vs_bane and 27m_vs_30m are not shown due to out-of-memory error on an NVIDIA Geforce RTX 3090 GPU.

ResQ/ResZ can obtain the close-to-optimal performance after 0.25 million steps in the 2s_vs_1sc scenario, shown in Fig. 3 (e). Although the other methods can learn close-to-optimal policy with more training steps, their performance is not stable. For the 1c3s5z scenario, depicted in Figure 2 (f), ResQ/ResZ are among the best-performing algorithms.

In the 2c_vs_64zg scenario, 2 Colossi agents fights against 64 enemy agents. It has a large action space, and the reward of each action varies rather than is constant as in other scenarios. As it is depicted in Fig. 3 (g), ResZ can obtain the best performance. This indicates ResZ's ability to model distributional reward. The results for 8m_vs_9m and 27m_vs_30m are depicted in Fig. 3 (h) and (f), respectively. ResQ and ResZ are the second-best performing algorithms. DMIX is good at scenarios which consist of homogeneous agents (e.g., 8m_vs_9m), but it under-performs for scenarios with heterogeneous agents (e.g., MMM2, 3s_vs_5z), where agent interactions are complex.

## 5.4 Predator Prey

In the predator prey environment [26], 8 agents hunt 8 preys in a $10 \times 10$ grid world. Each agent has 6 actions: moves in either 4 directions, stands still, and catches prey. If two adjacent agents execute simultaneously the *catch* action to a nearby prey, a reward $r = 10$ is given to the agents, but a failed catch by a single agent is punished by a reward $p \leq 0$. The more negative $p$ is, the higher level of coordination is needed for the agents. Following [8], the $\epsilon$ of the $\epsilon$-greedy strategy for QPlex, ResQ, and CW/OW QMIX is gradually annealed from 1 to 0.05 within 1 million steps.

The test return for three punishments: $p = 0$, $p = -2$, and $p = -4$ are shown in Fig. 4. For the most easy scenario $p = 0$, ResQ, QMIX, and QPlex can learn the optimal policies quickly. The performance of CW/OW QMIX drops in the middle of training. For the difficult $p = -2$ case, ResQ can obtain the best returns. CW QMIX and OW QMIX have the second and third best scores. QTRAN and QMIX can not obtain any positive returns. For $p = -4$, ResQ can obtain the best returns in the end. These results indicate ResQ's ability for challenging cooperation scenarios.

## 5.5 Ablation Study

We study the impact of different implementations for the main function and the residual function, and Starcraft versions on the SMAC benchmark. We find that *ResQ can improve the performance of multiple value factorization approaches through the use of the residual function $Q_r$.*

**Main Functions** In SMAC, QMIX is used as the implementation of $Q_{tot}$. For ResQ, the results of using QMIX, QAtten, and VDN as the main function $Q_{tot}$ are depicted as ResQ, Qtot-QAtten, Qtot-VDN in Fig. 5 (a) and (b). As it is shown, for the MMM2 and 8m_vs_9m scenarios, ResQ, Qtot-QAtten, and Qtot-VDN perform better than QMIX, QAtten, and VDN, respectively. This indicates that ResQ can *improve* the performance of value factorization approaches (the main function) through the use of residual functions and mask. Moreover, we find that QMIX is the preferred main function for the SMAC benchmark.

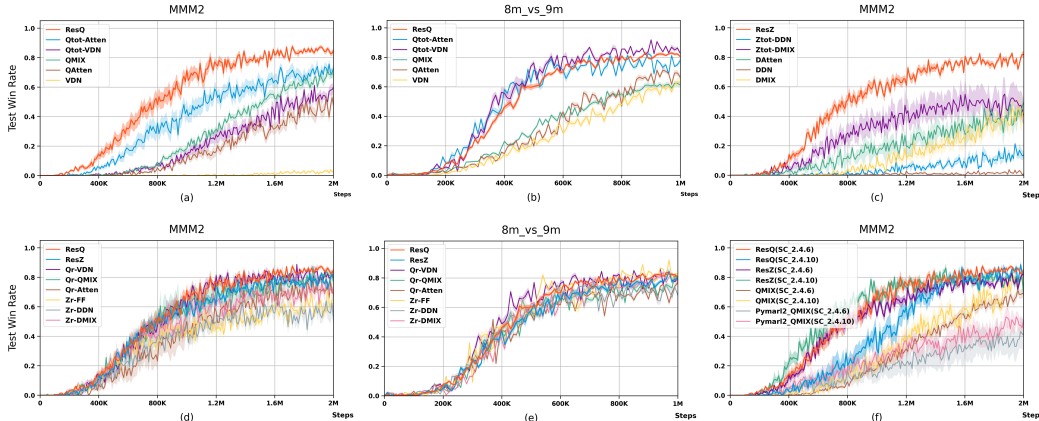

Figure 5: Impacts of the main functions (a), (b), (c), the residual functions (d) (e), and StarCraft versions (f)

For the distributional case, shown in Fig. 5 (c). The results of using DAtten, DDN, and DMIX as the main function $Z_{tot}$ are depicted as ResZ, Ztot-DDN, and Ztot-DMIX. DAtten is designed by us; it integrates IQN [10] with QAtten. ResZ, Ztot-DDN, and Ztot-DMIX perform better than DAtten, DDN, and DMIX, respectively. This indicates that ResZ can *improve* the performance of stochastic value factorization methods through residual functions.

**Residual Functions** The results of using different mixers (factorization methods) for the residual functions are shown in the Fig. 5 (d) and (e). The performance of using a feed-forward network (FF) as $Q_r$ (denoted as ResQ) is similar to that of using advanced mixers such as QMIX (Qr-QMIX) and VDN (Qr-VDN), and higher than that of QAtten (Qr-QAtten). In ResQ, we do not restrict the function classes of $Q_r$. This suggests that using a simple feed-forward network may be sufficient for $Q_r$. For the distributional case, the performance of using DAtten, FF, DDN, DMIX is depicted as ResZ, Zr-FF, Zr-DDN, Zr-DMIX. As it is depicted in Fig. 5 (e), we find that using DAtten as the residual function $Z_r$ is preferred.

**The Starcraft II version** used in this work is SC2.4.6.2.69232 (2.4.6 for short), which is the same version used as QMIX [4] and WQMIX [8]. Some methods (e.g.,[7]) uses the version SC2.4.10. We compare ResQ with Pymarl2_QMIX [14], which is a fine-tuned QMIX, across two versions. Figure 5 (f) depicts the results for different methods. The version of StarCraft is written inside the parentheses after the name of each method. For example, ResQ (2.4.6) denotes runing ResQ against SC2.4.6. As it is shows that ResQ/ResZ perform better than QMIX and Pymarl2_QMIX for both two versions in the MMM2 scenario. As Pymarl2_QMIX performs better than most algorithms in most case for SC2.4.10 [14], this suggests that ResQ/ResZ may perform better than other algorithms in SC2.4.10.

## 6   Conclusion

In this work, we propose, ResQ, a residual function-based approach for Multi-Agent Reinforcement Learning (MARL) value function factorization. ResQ recovers the optimal policy for any joint state-action value function by masking out state-action value pairs from the value function. We show that ResQ can satisfy the individual-global-max (IGM) principle and the distributional IGM principle without representation limitations. Through extensive experiments on multiple MARL tasks, we show that ResQ can obtain promising results.

**Acknowledgement** This work was partially supported by the National Natural Science Foundation of China (61872376, 61972409), by open fund of PDL (WDZC20215250113), by the China Post-doctoral Science Foundation (No.2021M690094); the FuXiaQuan National Independent Innovation Demonstration Zone Collaborative Innovation Platform (No.3502ZCQXT2021003). We would like thank the anonymous reviewers for their valuable comments.

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
