# OpenReview forum: "ResQ: A Residual Q Function-based Approach for Multi-Agent Reinforcement Learning Value Factorization"
_NeurIPS.cc/2022/Conference — NeurIPS 2022 Accept_

### Official Review · Reviewer_dP9n · 2022-07-08

**Rating:** 4
**Confidence:** 4
**Soundness:** 3 good
**Presentation:** 3 good
**Contribution:** 2 fair

**Summary:**

The paper proposed ResQ, a value factorization method for converting the joint Q function to a monotonic and residual part. Theoretical results show that the proposed method can satisfy the IGM condition, and experimental results show that it performs well in the matrix game, SMAC, and Predator Prey.

**Questions:**

- ResQ factorize the joint Q into the monotonic part and residual part, whereas Qtran factorize the joint Q into the monotonic part and residual part as well. The difference between ResQ and Qtran is that Qtran does not explicitly learn a residual function; instead, it employs an equality and an inequality which work similar to the coefficient w in ResQ. So, what is the benefit of learning the residual function? Does it aid in the reduction of approximation errors or the improvement of sample efficiency?
- Both Qtran and ResQ learns a joint Q network and trains individual Qs through another MSE loss. One of the main issues of Qtran is that the IGM is not guaranteed when the loss is not well minimized, which may lead to poor results in complex tasks. Is ResQ affect by the same problem? By the way, in the submitted code, Q_r is trained using loss max(Q_r, 0), which differs from that described in the paper.
- In matrix game, why is the Q_jt of ResQ(table 2(b)) and Qtran (table 2(f)) so different from the payoff matrix? They are expected to be the same as the payoff matrix (like the Q_jt in Qplex) because they are both trained using MSE(reward,Q_jt) with no expressive constraints. Moreover, I believe that training the network to distinguish 0.001 is inapproproate. Because of the learning rate and stochastic gradient descent, the values are likely to differ by more than 0.001 each iteration after convergence.


**Strengths And Weaknesses:**

The paper is written clearly in general, but it could be better organized to bridge the expectational and distributional parts. The paper investigates an important issue of value factorization methods, and the experiments appear to be adequate. The main weakness of this paper, in my opinion, is that the factorization and loss, both of which are similar to Qtran, have no better theoretical guarantees than Qtran and Qplex. The paper does not go into detail on how ResQ achieves "low approximation errors and high sample efficiency." It is unclear why ResQ performs better.

---

> ### Author Response · Authors · 2022-08-02
> **Response to Reviewer 4 (part 1)**
>
> We thank the reviewer for your time and effort in reviewing this submission. The valuable comments can help us significantly improve the quality of this work. We have added more experimental results to address the concerns of the reviewer in the appendix of the new version of this work, and we will answer the reviewer's comments as follows.
>
> **Comparison to QTran and QPLEX**
> ResQ can be viewed as a generalization of QTran, QPLEX, and  CW/OW QMIX, and it has a better theoretical guarantee than QTran, QPlex, and CW/OW QMIX. ResZ extends ResQ from expectation-based RL to distributional RL. It can be viewed as a generalization of DMIX and DDN. We show in this work that using the residual function for factorization can satisfy both the IGM and the DIGM theorem. Formula (8) in Theorem 2 ensures that ResQ can find a state-action value function which models all sub-optimal state-action pairs well. This property makes ResQ can achieve a lower approximation error than QTran. Please refer to the Main Response for a more comprehensive comparison to QTran, QPlex, and CW/OW QMIX.
>
> **Why does ResQ perform better**
> To answer this question, we have evaluated the performance of 10 variants of ResQ/ResZ. We have studied why ResQ performs better in Figure 5 of the original submission. ResQ is a combination of the main function, the mask, and the residual function. In default, ResQ uses QMIX as the main function. It can be viewed as the combination of QMIX with the mask and the residual function. We have shown in Figure 5 (a) and (b), ResQ performs much better than QMIX. Further, we have studied two variants of ResQ: Qtot-Atten and Qtot-VDN. They use QAtten and VDN as the main function instead of QMIX. The experimental results in Figure 5 (a) and (b) show that Qtot-Atten and Qtot-VDN perform better than QAtten and VDN, respectively. This indicates that with the use of the residual function and the mask function, the performance of the original value factorization methods (VDN, QAtten, QMIX) can be improved.
>
> We have studied why ResZ performs better in the original submission in Figure 5 (c). ResZ can be viewed as the combination of DAtten with the mask and the residual part. DAtten is a stochastic variant of QAtten. We find that ResZ performs better than DAtten for the MMM2 scenario. And we have studied two variants of ResZ: Ztot-DDN and Ztot-DMIX. They use DDN and DMIX as their main function, respectively. We find that Ztot-DDN and Ztot-DMIX perform better than DDN and DMIX, respectively. This indicates that for the stochastic case, using the residual functions and the mask can improve the original value factorization methods.
>
> Figure 5 (a) (b) (c) shows that using the residual functions and the masks can improve the performance of existing value factorization methods. We study whether the neural networks of the residual functions affect the ResQ significantly. The implementations of the residual functions we consider are VDN, QMIX, QAtten, feed-forward, DDN, and DMIX. As it is depicted in Figure 5 (d) and (e), for the MMM2 and the 8m\_vs\_9m scenarios, using a simple feed-forward (FF) network is enough for the expectation-based RL. For the stochastic case, using DAtten is preferred.
>
> **Questions**
>
> *Reply to Q1*
>
> To study the reason why ResQ performs better than QTran, We use two variants of ResQ: Qtot-VDN and Qtot-VDN-MSE. Qtot-VDN uses VDN as the main function same as QTran. The difference between Qtot-VDN between Qtran is the residual function, the mask, and the inequality conditions. Qtot-VDN requires $Q_r \leq 0$ and QTran requires that $Q_{tran}(\tau, u) \ge Q(\tau, u) \quad \forall u \neq \bar{u}$. Qtot-VDN-MSE uses MSE loss to implement $Q_r \leq 0$, and it uses VDN as its main function. For MMM2, Qtot-VDN can obtain a win rate of 0.6 (see Figure 5 (a)). The win rate for QTran is 0 (see Figure 3 (a)). For the 8m\_vs\_9m scenario, Qtot-VDN can obtain a win rate of 0.8 (see Figure 5 (h)). And QTran can obtain a win rate of 0.4. The performance gap does not come from the implementation of the main function; it comes from the residual function, the mask function, and the inequality conditions. Further, Qtot-VDN-MSE performs similarly to Qtot-VDN, and it performs better than QTran. This suggests that the performance gap between ResQ and QTran does not come from the implementation of inequality conditions for ResQ.

---

> > ### Author Response · Authors · 2022-08-02
> > **Response to Reviewer 4 (part 2)**
> >
> > *Reply to Q2*
> >
> > ResQ does not significantly affect by inequality violations. The residual function uses the negative absolute function to ensure that $Q_r \leq 0$ (Please refer to lines 216-217 of rest\_q\_learner\_central.py). We have evaluated the performance of a variant of ResQ, ResQ-MSE, which uses MSE loss to implement $Q_r \leq 0$. We have studied the performance of ResQ-MSE in the MMM2, MMM, and 8m\_vs\_9m scenarios. The performance of ResQ-MSE and ResQ are similar in these three scenarios. This indicates that the violation of inequality conditions does not affect the performance significantly.
> >
> > We agree with the reviewer that using the MSE loss in QTran to ensure the inequality constraints could lead to violations of the IGM principle. We have studied QTran's MSE loss caused by inequality violations in the MMM2 scenario. The MSE loss is 2e-3, which suggests that only a few inequality violations exist.
> >
> > *Reply to Q3*
> >
> > For Table 2, the optimal value estimated by ResQ is 8.10, which is 0.1 difference from 8.00, the true optimal value. ResQ has the best approximation quality regarding the optimal action. QTran and QPlex fail to find the optimal action. The wrongly estimated optimal value of QPlex is 7.49. Its gap to the optimal value (0.51) is much larger than the gap of ResQ (0.1). For sub-optimal action values in Table 2, QPlex indeed recovers its values more closely than ResQ and QTran. However, For the matrix listed in Table 1 of the appendix, QPLEX does not model more closely the sub-optimal actions than ResQ/ResZ and QTran.
> >
> > In general, QPlex does not perform well when recovering the optimal actions for difficult pay-off matrices. We have studied three more matrices to evaluate all the studied algorithms. For the matrix [[2.5, 0, -100],[0, 2, 0],[-100, -100, 3]], Only ResQ and QTran can recover the optimal action, the other algorithms cannot. For the matrix [[8, -12, -12],[-12, 0, 0],[-12, 0, 7.9]], Only ResQ, CW QMIX and QTran can recover the optimal action. QPlex does not model the sub-optimal actions better than ResQ and QTran. For the matrix  [[8, -12, -120], [-12, 7.8, 7.7], [-120, -130, 7.9]], ResQ recovers the optimal action, and the other algorithms do not.
> >
> > We choose 7.999 to emphasize that QTran is likely to over-estimate sub-optimal state-action pairs. QTran wrongly estimated the optimal action and enlarged the value gap. The gap estimated by QTran between the first and the second largest value is 0.02, which is larger than the gap (0.001) between 8 and 7.999. We agree with the reviewer that 0.001 difference may be too small for RL algorithms due to their stochastic nature. We will replace 7.999 with 7.9, and add all the new experimental results to the paper.

---

> > > ### Comment · Reviewer_dP9n · 2022-08-03
> > > **Response to the authors**
> > >
> > > I would like to thank the authors for their feedback. However, it does not address my major concerns, which I raised following the Main Response.
> > >
> > > Furthermore, in the matrix game, I’m still puzzled as to why the $Q_{jt}$ of ResQ and Qtran have such a large gap to the payoff matrix. They are both trained using $MSE(r, Q_{jt})$ with sufficient exploration and no  representation limitation. Therefore, it is a very simple regression problem, and the $Q_{jt}(u_1,u_2)$ should be very close to $r(u_1, u_2)$ for any action pair. Is there something I’m missing?

---

> > > > ### Author Response · Authors · 2022-08-06
> > > > **Response to Reviewer (both QTran and ResQ fit the simple regression problem well)**
> > > >
> > > > $Q_{jt}$ of ResQ is $Q_{tot} + w_r Q_r$ which is used to recover the optimal actions. The $Q_{jt}$ of QTran (We will write it as $Q_{tran}$ following the definition of the QTran++ paper)  is defined as $\sum_{i=1}^n Q_i + V(\tau)$; it is used to recover the true optimal action as well. We will make this clear in the new version of this work to avoid confusion. The matrices shown in the paper is used to demonstrate the ability to recover optimal policies rather than the quality of reconstruting Q values.
> > > >
> > > > $Q_{tran}$ of QTran is not used to approximate the true value function, and it is stated in the QTran paper *"We found that condition (4b) is often too loose, leading the neural networks to fail their mission of constructing the correct factors of Qjt"*.
> > > >
> > > > Both ResQ and QTran use the unrestricted value function $Q_{appro}$ to approximate $Q$, and then they learn $Q_{jt}$ to approximate the optimal policy of $Q_{appro}$. We agree with the reviewer that  approximating $Q_{appro}$ to $Q$ is a simple regression problem. $Q_{appro}$ approximates $Q$ well for both ResQ and QTran. Due to limited page space, we do not show the matrix of $Q_{appro}$ in the submitted paper. We have run ResQ for longer timesteps (from 20,000 to 50,000), the $Q_{jt}$ of ResQ approximates $Q$ well, but $Q_{tran}$ does not.
> > > >
> > > > We will list $Q_{tot}$, $Q_r$, $Q_{tran}$, $Q_{appro}$, and $Q_i$ in the appendix of the paper.

---

> > > > ### Author Response · Authors · 2022-08-09
> > > > **Response the Reviewer dP9n**
> > > >
> > > > We would like to express our gratitute to the reviewer for this comments which help us improve the quality of this work.
> > > >
> > > > We have changed the sentence *"Achieving the IGM and the DIGM principles with low approximation errors and high sample efficiency remains an open challenge"* to be *"Achieving the IGM and the DIGM principles without representation limitations remains an open challenge"*.
> > > >
> > > > We have updated Figure 1 (from 7.999 to 7.9) and updated Table 2 (from 20,000 to 50,000 steps); we have update the distributional matrix (from 7.999 to 7.9 as well) and present the $Q_{jt}$ of all the methods.
> > > >
> > > > We have added one more table in the experimental section to describe the $Q_{tot}$, $Q_{jt}$, $Q_i$, $Q_{tran}$ and $Q_r$ of different methods.
> > > >
> > > > We have changed the description regarding QTran via removing *"$Q_{tran}$ over-estimates sub-optimal actions"* from the paper.

---

### Official Review · Reviewer_rgjE · 2022-07-10

**Rating:** 7
**Confidence:** 2
**Soundness:** 3 good
**Presentation:** 2 fair
**Contribution:** 3 good

**Summary:**

This paper extends distributional QMIX with a residual value function. Distributional QMIX (DMIX) an multi-agent reinforcement learning (MARL) algorithm using distributional Q-function to tackle stochasticity in MARL. DMIX decomposes the joint Q-value function of multiple agents into several sub value functions and aggregate them with a so-called mixer network. The limitation of DMIX is that each sub Q-function needs to be monotonically increasing with respect to the joint Q-function (i.e., the derivative of the joint Q-value w.r.t., the sub Q-value must be non-negative. This paper removes this limitation by introducing a residual Q-function in the factorization. The experimental results show that the proposed method improves the performance over the baselines.

**Questions:**

- Could the author comment on a concurrent work [1]?
- Do the boldsymbols stand for vectors? If so, why $Q_{jt}$ takes vector of actions as inputs in Equation (1) while taking scalar actions as inputs in Equation (4)?
- As the proposed method uses more parameters (because of residual network) than DMIX, did the author try comparing with DMIX in a larger architecture?

[1] Residual Q-Networks for Value Function Factorizing in Multi-Agent Reinforcement Learning, https://arxiv.org/abs/2205.15245

**Limitations:**

I encourage the author to think of the limitation of representation limits of residual decomposition. For example, could this residual decomposition be able to account all kinds of payoff functions?

**Strengths And Weaknesses:**

Originality: The use of residual decomposition is not new, yet this is the first time being used in factorizing value function in MARL. As such, I acknowledge the originality.

Quality and clarity:
The figure 1 keeps being cited in the paper to illustrate the representation limitations and difficulty of modeling the value function. Yet, I don't understand how this makes value function representations difficult. I would like to see further explanation from the author.

Significance:
This work lifted the monotonicity assumption in QMIX framework. It is remarkable to my knowledge since non-monotonic payoff functions could exist in some applications.

---

> ### Author Response · Authors · 2022-07-31
> **Response to Reviewer 3**
>
> We want to thank the reviewer for the effort and time in reviewing this paper. We will take your valuable feedback into account to improve our work.
>
> **Non-monotonic matrix**
> The matrix shown in Figure 1 is a complicated pay-off matrix (denoted as Matrix A). It is more difficult than the popular non-monotonic pay-off matrix (denoted as Matrix B). Matrix B is proposed in QTran, and its array form is [[8, -12, -12], [-12, 0, 0], [-12, 0, 0]]. This is a non-monotonic matrix. In Matrix B, there are two agents: agent 1 and agent 2. The optimal action is choosing the first action for both agents. If we let agent 2 chooses the first action, the reward vector for agent 1 becomes [8, -12, -12]. For agent 1, the reward monotonically increases from the bottom to the top (the value increases from -12 to 8). If agent 2 chooses the second/third action, the rewards for agent 1 become [-12, 0, 0]. The reward monotonically increases from the top to the bottom. Note that the direction of monotonic increment differs between the first action and the second/third actions. QMIX, VDN, DMIX, and DDN cannot learn such a non-monotonic pay-off matrix due to their representation limitations.
>
> Matrix A (the one depicted in Figure 1 of this work) is a more difficult non-monotonic matrix than Matrix B. The array form of Matrix A is [[8, -12, -12], [-12, 0, 0], [-12, 0, 7.999]]. For matrix A, if agent 1 chooses the first action, the reward vector for agent 2 becomes [8, -12, -12]. It monotonically increases from the right to the left. If agent 1 chooses the second/third action, the rewards ([-12, 0, 0] and [-12, 0, 7.99]) for agent 2 monotonically increase from the left to the right. The direction of increment is different between the first and the second/third action. If agent 2 chooses the first action, the rewards for agent 1 become [8, -12, -12]. It increases from the bottom to the top. If agent 2 chooses the second action, the rewards for agent 1 become [-12, 0, 0]. The reward increases from the top to the bottom. If agent 2 chooses the third action, the rewards for agent 1 become [-12, 0, 7.999]. The reward does not increase from the bottom to the top. Matrix A is a more challenging pay-off matrix for MARL value factorization than Matrix B. QTran, QPLEX, CW QMIX, OW QMIX cannot model this matrix well.
>
> **Limitations**
> In this work, we assume that the approximated argmax operator can find the optimal actions, however, this does not always true. For scenarios with multiple agents and a large action space, the possibility of executing the correct optimal action decreases exponentially with the increase of agent count. In these scenarios, the argmax operator may fail. We have shown that with the perfect argmax operator, ResQ can find the optimal policy for tabular state-action values. If we approximate the state-action values, state with neural networks, the neural network approximation error could interact with the error of the argmax operator. In these scenarios, ResQ may fail.
>
> **Question**
> 1. The concurrent work, Residual-Q-Network (RQN), deals with MARL value factorization too. It extends QTran by adding an individual correction factor for each utility function to compute an adjusted utility function (individual Q value). And the sum of the adjust utility function is used as the transformed Q value $Q_{tran}$ which is defined in QTran. RQN can be viewed as a special case of ResQ. It uses the sum of per-agent utility as the main function $Q_{tot}$, and the sum of individual correction factors as the residual function $Q_r$. In this work, $Q_{tran}$ over-estimates the value of sub-optimal actions (see 6b of Theorem in the RQN paper). RQN uses the word “residual” because they implement the adjusted utility function in a way similar to ResNet (Deep Residual Learning for Image Recognition, CVPR 2015). ResQ uses the word “residual” to indicate the mask-out values from the Q function. We will compare ResQ and RQN in the related work.
>
> 2.
> The bold symbols are used in Section 2; they are used to represent vectors. We will change all the notation of vectors using bold symbols to make the presentation of this work better.  $Q_{jt}(\boldsymbol{\tau}, \boldsymbol{u})$ takes a vector of actions (i.e., \boldsymbol{u}) as its input. And $Q_i(\tau_i, u_i)$ take the action $u_i$ as input.
>
> 3.
> By modifying the dimension of the last layer of the hypernet inside DMIX, we have increased the number of parameters of DMIX from 85K to 350K, which is bigger than that of ResQ (319K) and that of ResZ(316K). Let's denote DMIX with more parameters as DMIX-larger, and test its performance in the MMM2, the MMM, and the 3s\_vs\_5z scenarios.  DMIX-larger performs slightly better than DMIX in the MMM2 scenario but worse than DMIX in the MMM and the 3s\_vs\_5z scenarios. ResQ and ResZ performs better than DMIX and DMIX-larger in these scenarios. This means that the performance improvement of ResQ over DMIX does not come from the use of more parameters.

---

### Official Review · Reviewer_Ea2R · 2022-07-16

**Rating:** 7
**Confidence:** 4
**Soundness:** 3 good
**Presentation:** 2 fair
**Contribution:** 3 good

**Summary:**

Authors introduce a constraint-free residual function into the state-action value function factorization construct, thereby allowing the joint (main, residual) function to be able to express a larger class of functions than with the main functions alone. The proposed algorithm is tested against a rich set of baselines and benchmarks and shows substantial improvements.

**Questions:**

Q1: continued from the Strengths and Weaknesses: how much complexity has been added with the ResQ function approximator NN?

Q2: How would ResQ be positioned in the literature against, for example, MAVEN? MAVEN had its take on the different classes of state-action joint value function factorization with the corresponding algorithms published at the time. How would ResQ's take differ in terms of representational complexity?

Q3: Under what circumstances do the approximations for the mask functions fail? What assumptions, if any, should hold in order for the approximations to remain valid? How strong would you say are those assumptions?

**Limitations:**

Please do include some analyses on the computational costs of ResQ

**Strengths And Weaknesses:**

The paper is easy to follow, and the claims made in the abstract are adequately addressed, with relevant theoretical setup, literature review, and experiment results analyses. Findings from the experiments indicate that the proposed algorithm works well and explainable MARL settings such as the matrix game make the better performance attributable indeed back to the motivating example.

One potential shortcoming of the paper could be the lack of insights provided on the computational costs incurred in calculating the residual functions. As ResQ requires fully connected neural networks to estimate the residual functions, one training step of ResQ would probably be more expensive than that of other baselines. Some analyses into this matter would be a nice addition to the draft.

---

> ### Author Response · Authors · 2022-07-31
> **Response to Reviewer 2**
>
> We agree with the reviewer that learning a residual function indeed incurs more computation than QMIX and VDN, but it requires a similar cost as QTran and QPlex. If we implement the residual function the same as the main function, the inference/back-propagation cost for the whole neural network could be twice the cost of using the main function alone during the training of the value function. After the value function is trained, each agent executes its action greedily according to its utility function. During agent execution, the inference cost for all the algorithms is the same. We will add a discussion of the computation cost of ResQ into the new version of this work.
>
> ResQ needs a similar computational cost for inference/back-propagation as QTran and QPlex. QTran and QPlex learn $V$ functions to assist the learning of the state-action value function. The $V$ function is used to assist the implementation of the IGM principle, and its output is a scalar value. The input of QTran's $V$ consists of state only. And the input of the $V$ function of QPlex consists of state and utility values. The input of ResQ's $Q_r$ is the same as QPlex. From the view of the input and the output for these functions (V and residual functions), they need a similar computational cost for neural network interference and back-propagation.
>
> 1. Reply to Q1. Regarding the complexity, we have discussed the time complexity of ResQ. For the space complexity, ResQ does not require much more parameters than other algorithms. For the MMM2 scenario of the SMAC benchmark, the number of neural network parameters of value functions for ResQ, ResZ, ResZ-DMIX, QPlex, QMIX, and DMIX are 319K, 316K, 331K, 342K, 85K, and 85K, respectively. ResQ requires a similar parameter size as other algorithms, which do not suffer from representation limitation issues. Albeit consuming fewer neural network parameters, QMIX and DMIX have representation limitation issues. For implementation complexity, we use a simple feed-forward network to implement the residual function, which is quite straightforward.
>
> 2. Reply to Q2. We believe that the reviewer is referring to the paper “Mahajan et al. MAVEN: Multi-Agent Variational Exploration. NeurIPS 2019”. MAVEN deals with the inefficient exploration problem in MARL. Inefficient exploration problems could interact with the representation limitation problems. As it is stated in the MAVEN paper, this may push the algorithm towards sub-optimal policies. In MAVEN, value-based agents condition their behaviour based on a shared latent variable. And the latent variable is controlled by a hierarchical policy. MAVEN relies on the underlying value factorization method to decompose Q values. And it adopts QMIX as its value factorization method. ResQ can be used in MAVEN as its factorization method. We will compare ResQ with MAVEN in the related work in the new version of this work.
>
> 3. Reply to Q3. Knowing the optimal action over Q is computationally intractable, we make approximations to derive a practical algorithm. For a scenario requiring highly-coordinated agent exploration, it is difficult for ResQ to find the optimal actions. The approximated mask function will fail if the approximated optimal action differs significantly from the optimal action. In ResQ, we assume that the approximation argmax of $Q_{tot}$ can lead to correct optimal actions. However, this assumption does not always hold. We think that combining MAVEN with ResQ can make the mask function more robust than using ResQ alone. We will discuss the limitation of this work in a discussion section of the new version.

---

### Official Review · Reviewer_HPe6 · 2022-07-23

**Rating:** 7
**Confidence:** 4
**Soundness:** 3 good
**Presentation:** 3 good
**Contribution:** 3 good

**Summary:**

Post rebuttal:
> I am updating my score post rebuttal based on the responses provided by the authors. They have answered almost all my questions satisfactorily. My increased score is contingent on the authors making the appropriate modifications to their paper as mentioned in their rebuttal.


This paper extends the action-value function factorization based approaches developed for centralized training and decentralized execution in cooperative multi-agent reinforcement learning problems. The main contribution is decomposing the action-value function into a factorizable component and a residual (formed by masking some state-action value pairs), such that the factorizable component alone is sufficient for each agent to independently choose their optimal actions, which also leads to joint optimality for all agents. The authors develop this method for both expected value and stochastic value (distributional) RL. The main challenges addressed by this method are improved representation capability, sample efficiency, and approximation error. The authors analyze their method theoretically and empirically through experiments on matrix games, predatory-prey and StarCraft benchmarks.

**Questions:**

Major comments:
1. Can the authors provide a detailed comparison with QTRAN and QPLEX which also appear to have a residual decomposition, where the residual is a value function? Is *ResQ* a generalization of QTRAN and QPLEX? What are the benefits of *ResQ* over these algorithms other than generalization?
2. I did not understand why *ResQ* can track joint state-action value pairs more closely than other algorithms. Can the authors please elaborate this point?
3. The authors mention that they restrict attention to discrete actions. Is this a fundamental limitation of the approach? What needs to be done to extend this work to continuous action spaces?
4. Can the authors explain why the joint state-action value function of the task shown in Fig.1 cannot be expressed well by monotonic increasing functions (Ref. Line 81, Lines 133-134)? This might be obvious, but one or two sentences explaining this would be helpful.
5. In the Related Work section, especially in the first paragraph, the authors list several related works. I think, it would be more useful to present these works in context to the work of the present paper instead of just listing them.
6. In Lines 124-126, the authors state: "QTRAN over-estimates sub-optimal actions, which may lead to non-optimal decisions, whereas ResQ can estimate correctly non-optimal state-action values." Isn't it shown in the QTRAN paper that under appropriate factorization, QTRAN recovers the joint optimal policy, despite this over-estimation? Can the authors clarify and elaborate on this comment in the paper? This is again mentioned in Lines 135-137, but a more detailed explanation would be helpful.
7.  In Line 459, in the Appendix, the authors write that "Proof. This Lemma was proved as Theorem 4 of [8]." for the Proof of Lemma 1. I could not find this theorem in [8]. As this Lemma is crucial for proving Theorem 2, I could not follow that proof as well. Can the authors provide an updated reference or am I missing something here?
8. In Fig.2, what is the input to the residual mixer in part (a)?
9. Doesn't approximating the mask function with the current best action ($\tilde u$) instead of the best action ($\bar u$) make the algorithm dependent on the initialization? Does the error introduced by this approximation go to 0 as the iterations progress? Is there any theoretical or empirical evidence for this?
10. Why ResZ was not tried for the predator-prey example?
11. Why is ResQ slower to learn in the p=-4 case in the predator prey example shown in Fig. 4?
12. What is the motivation behind factorizing the residual function $Q_r$ in the ablation study in Sec 5.5?

Minor comments:
1. In Line 67, the equation is written as $s_{t+1} \sim P(\cdot|s^t, \mathbfit{s^t})$. Shouldn't this be $s_{t+1} \sim P(\cdot|s^t, \mathbfit{u^t})$?
2. Can the $*$ superscript be defined in Line 70 for ease of comprehension for readers?
3. Shouldn't the policy in Line 70 be $\pi_i(u_i|\tau_i)$ instead of $\pi_i(u|\tau_i)$?
4. In line 71, shouldn't the joint policy be $\pi = \langle \pi_1, \dots, \pi_n \rangle$ instead of $\pi = \langle \pi_1, \dots, \pi_N \rangle$?
5. The joint action-observation history is defined as $\tau^N$ in line 70 and as $\mathcal{T}^N$ in line 75. Can this be made consistent throughout?
6. In line 76, consider replacing 'histories' with 'history'.
7. In Line 76, shouldn't it be $Q_i: \mathcal{T}_i \times \mathcal{U}_i \to \mathbb{R}$ rather than $Q_i: \mathcal{T} \times \mathcal{U} \to \mathbb{R}$?
8. In general *argmax* is a set operation and hence the output is a set of candidates. It would be good if the authors add a line stating that in this paper, they assume *argmax* is either unique or the ties are broken in a consistent manner so that equation (1) holds.
9. In equation (1), consider replacing $N$ with $n$ for consistency. I think $n$ is the index of the last agent, and $N$ is a superscript that denotes the set of all agents. I think this confusion appears in multiple locations in the paper (such as lines 78, 180, 181, 484 etc.). It would be good to rectify this.
10. It would be good to define $\mathcal{N}$ or $N$ as the set of agents as it is used explicitly in equation (2), and then this notation can be used consistently in the paper.
11. I could not understand this sentence in lines 88-90: "IQN updates its value distribution through a distributional Bellman operator, which ensures the value distribution follow the same distribution." Can this be rephrased?
12. I think the symbols should be $w_i, w_j$ instead of $w_1, w_2$ in Line 91.
13. In Line 134, it should be "factorize" instead of "factorized".
14. In Lines 147, 148, it should be "joint" instead of "jointed".
15. In Line 151, $\tau$ is referred to as an observation, where in fact, it is a joint action-observation history.
16. In Line 237, shouldn't it be "Table 2" instead of "Table 1a"?
17. In Line 454, shouldn't "From (2)" be after equation (5) and not equation (4)? And the reason for (4) should be $\bar u$ maximizes $Q_{tot}$?
18. $Q_r(\tau, u)$ is written in Line 497. Shouldn't it be $Z_r(\tau, u)$?
19. In Line 524, there is a typo: "matrxi" -> "matrix".








**Limitations:**

I think the authors need to add more discussion around the limitation of their approach. Specifically, can all action-value functions be factorized using the *ResQ* approach? They address this partially via Theorem 2, which seems to suggest that all action-value functions can be factorized using *ResQ*. If not, under what conditions does it fail? A discussion on this point would be useful.

**Strengths And Weaknesses:**

Strengths:
1. The paper is well written. The original contributions are highlighted clearly.
2. The *ResQ* insight is a neat and useful one and I think it extends the applicability of factorization based approaches to a larger set of problems.
3. The authors provide theoretical analyses and also perform a detailed experimental study establishing the utility of their approach and improvement over the current best approaches.

Weaknesses:
1. Most of the theoretical concepts used in the paper such as IGM, DIGM have been already established in literature (and duly acknowledged by the authors). Furthermore, as far as I understand, the residual decomposition technique is very similar to QTRAN, QPLEX and the main difference, as far as I can see, is that it generalizes the decomposition proposed in QTRAN, QPLEX. As a result, most of the proofs in the current paper are also similar to QTRAN. Is my understanding correct?

---

> ### Author Response · Authors · 2022-08-01
> **Response to Reviewer 1 (part 1)**
>
> We want to express our sincere gratitude to the reviewer. Such a thoughtful and in-depth review can help us greatly improve the quality of this work. We will improve this work based on the comments.
>
> **Weakness**
> The proof of Theorem 1 has some similarities with QTran, but ResQ has a better theoretical guarantee than QTran. The proof of Theorem 2 is not similar to QTran. The proofs of Theorem 3 and 4 have some similarities with QTran, but they extend the idea of using residual functions from the expectation-based RL to expectation-based RL.
>
> **Majors comments**
>
> 1. We have compared ResQ with QTran and QPlex in the main response.
>
> 2. We have answered why ResQ can track value better than QTran and QPLEX in the main response. For CW QMIX and OW QMIX, they focus on modeling the value of the optimal actions but pay less effort to model the value of the sub-optimal actions. Thus, they do not model the value of sub-optimal actions well. We will soften the claim as "Compared to QTran, QPLEX, and weighted QMIX, ResQ could track joint state-action value pairs more closely.
>
> 3. ResQ works in the discrete-action domain. For the continuous-action domain, a new theorem analogous to the IGM theorem should be developed. And the mask function should be improved to mask out ranges of actions rather than a set of actions. We think it is promising to combine ResQ with MADDPG for the continuous-action domain.
>
> 4. In a two-agent non-monotonic matrix, if one agent selects different actions, the direction of reward increment becomes different for another agent. Let's take the matrix [[8, -12, -12], [-12, 0, 0], [-12, 0, 7.999]] as an example. If agent 1 chooses the first action, the reward vector for agent 2 becomes [8, -12, -12]. The reward for agent 2 monotonically increases from the right to the left. If agent 1 chooses the third action, the reward vector for agent 2 becomes [-12, 0, 7.99]. It monotonically increases from the left to the right.
>
> 5. Thanks, we will discuss them in the context of this work.
>
> 6. We have explained them in the main response.
>
> 7. Sorry for the mistake. It should be Theorem 1 of [8]. Theorem 1 of [8] shows that Weighted QMIX can always find a monotonically increasing function $Q_{tot}$ that shares the same optimal policy as the true value function of $Q(s,u)$. This indicates that for any $Q(s, u)$, there exists a monotonically increasing function $Q_{tot}(s, u)$ that shares the same optimal policy as $Q(s, u)$. We will improve the description of this lemma in the new version.
>
> 8. The input to the residual mixer consists of state, per-agent utilities, and actions.
>
> 9. Yes, the performance of approximated argmax operator depends on the initialization. The error induced by approximation gradually reduces during training. For example, the error of approximated argmax for the pay-off matrices in Table 2 and Table 1 (in the appendix) reduce to 0 in the end. However, for complex tasks such as the MMM2 scenario in the SMAC benchmark, we have observed from the game video that agents choose the wrong actions during testing, which leads to a game loss.
>
>
> 10. According to the reviewer's suggestion, we have run the three stochastic RL algorithms: ResZ, DMIX, and DDN on the predator-prey benchmark. For the scenario without punishment, these three algorithms perform similarly. ResZ is more stable than DMIX. However, for scenarios with punishment, none of the algorithms can obtain a decent policy that obtains positive rewards. These distributional RL algorithms are susceptible to the relative over-generalization pathology. However, the expectation-based version, ResQ does not suffer from this problem.
>
>
> 11. OW QMIX performed the best in the p=4 case, but it performed the worst in the p=0 case, and the second worst in the p=2 case. CW QMIX learned slightly faster than ResQ in the p=4 case. ResQ is the third best-performing algorithm in the p=4 case, and its performance is among the best-performing algorithms in the p=0 and p=2 cases. ResQ performs better than OW QMIX in the SMAC benchmark and matrix games all the time. We think because of the randomness of these RL algorithms and the environments, ResQ does not learn faster than OW QMIX in the p=4 case.
>
> 12. Figure 5 (a) (b) (c) shows that using the residual functions and the masks can improve the performance of the existing value factorization method. The input to the residual function consists of state, per-agent utilities, and actions. And it output a scalar value.  A value factorization function (mixer function) takes state, per-agent utilities, and actions as input,  and they output a scalar value too. We implement the residual function using these mixer functions to build on previous well-designed mixer functions. And then, we study the performance of different residual functions in the ablation study.

---

> > ### Author Response · Authors · 2022-08-01
> > **Response to Reviewer 1 (part 2)**
> >
> > **Minors**
> > 1. Yes, it is a typo.
> > 2. The symbol "*" represents 0 to T, where T denotes the time step. * means that the history could be short or long. For example, $\tau_1$ could $\in O^1_1\times U^1_1\times O^2_1\times U^2_1$ and could $\in O^1_1\times U^1_1$, where the superscript represent time, and the subscript is the index of a agent.
> > 3. Yes, it is a typo, we will fix it.
> > 4. Yes, we will make all the symbol more consistent.
> > 5. We would like to thanks the reviewer’s effort for reviewing this work. Yes, we will make all the symbol more consistent.
> > 6. We will replace “histories” as “history”
> > 7. Yes.
> > 8. Thanks for your suggestion. We will state in this work ``we assume the argmax operator is unique, the action with smallest index is selected to break ties if a tie exists''
> > 9. Yes, the symbol n represents the index of an agent.
> > 10. Thanks, we will use $\set{N}$ to represent set of agents in a more consistent manner.
> > 11. We have rephrased it as ``Distributional RL models full return distribution $Z(\tau, u)$ instead of $Q(\tau,u)$. IQN defined a distributional Bellman operator, and use it to updates its $Z(\tau,u)$. After applying the distributional Bellman operator on $Z(\tau,u)$, its resulting $Z(\tau',u')$ remains in the same distribution as $Z(\tau,u)$.''.
> > 12. Yes, we have fixed the typo in the new version.
> > 13. We have made “hard-to-factorize” consistent over the new version.
> > 14. Yes, it should be “joint” rather than “jointed”
> > 15. Yes, it is an action-observation history.
> > 16. Yes, it should be Table 2 a).
> > 17. Yes, the “from (2)” in formula (4) should be $\bar{u}$ maximize $Q_{tot}$, and “from (2)” should be after (5).
> > 18. Yes it should be $Z_r(\tau, u)$ instead of $Q_r(\tau, u)$
> > 19. Yes, it should be “matrix”.

---

### Author Response · Authors · 2022-08-01
**Main Response**

We faithfully thank all reviewers for their insightful comments and valuable feedback. The reviewers acknowledge the novelty and originality (R1, R2, R3), the significance (R1, R2, R3), good writing quality (R1, R2, R4), promising experimental results (R1, R2, R3, R4), and the theoretical contribution (R1, R2). We will incorporate the suggestions and address the concerns in the new version of this work. We have conducted 11 more experiments to address the comments, and their results are included in the appendix of the new version.  We will answer two common questions raised by the reviewers in this main response and answer other questions in a dedicated response to each reviewer.


**Comparison to QTran, QPlex, and CW/OW QMIX**
Besides satisfying the IGM and the DIGM theorem, ResQ can be viewed as a generalization of QTran, QPlex, CW/OW QMIX, and ResQ has a theoretical advantage over them.

We can reformulate QTran in the form of ResQ. QTran approximates the true value function $Q(\tau, u)$ via $Q_{tran}(\tau, u)$. $Q_{tran}(\tau, u) = Q_{tot}(\tau, u) + w_r(\tau, u)V(\tau)$, where $Q_{tot}(\tau, u) = \sum_i Q_i(\tau_i, u_i)$ and $w_r(\tau, u)=1$. And it must satisfy the inequity condition from 4b of Theorem 1 of QTran, which requires that $Q_{tran}(\tau, u) \ge Q(\tau, u)$ for all non-optimal actions. Thus, QTran could over-estimate state-action value pairs.

ResQ learns $Q_{jt}(\tau, u)$ to approximate the state-action value function $Q(\tau, u)$. Theorem 1 in ResQ shows that the state-action value function $Q_{jt}(\tau, u)$ satisfy $Q_{jt}(\tau, \bar{u}) \ge Q_{jt}(\tau, u)$, where $\bar{u} = [\bar{u_i}]^n_{i=1} \quad \bar{u_i}=$ arg$\max_{u_i}Q_i(\tau_i, u_i)$.  Further, as it is shown in Formula 8 of Theorem 2 in ResQ, $Q_{jt}(\tau, u) = Q(\tau, u) \quad \forall u \neq \bar{u}$, ResQ can find a $Q_{jt}(\tau, u)$ that matches the sub-optimal state-actions of $Q(\tau, u)$ closely. QTran cannot guarantee that the learned approximated function $Q_{tran}$ satisfies this property. Further, ResQ uses $Q_r(\tau, u)$ to model the residual part; it has more input and is more flexible than the residual part $V(\tau)$ of QTran.

ResQ can be viewed as a generalization of QPlex as well. According to QPlex, QPlex learns a function $Q_{plex}(\tau, u)$ to approximate the state-action value function $Q(\tau, u)$.  $Q_{plex}(\tau, u) = V(\tau) + Adv(\tau, u)$, where $V(\tau) = max_u Q_{plex}(\tau,u)$. According to Formula (11) of the QPlex paper, we can rewrite $Q_{plex}$ in the form of ResQ as $Q_{plex}(\tau, u) = Q_{tot}(\tau, u) + w_r(\tau, u) Q_r(\tau, u)$, where $w_r(\tau, u) = 1$, $Q_{tot}(\tau, u)=\sum_i Q_i(\tau, u_i)$ and $Q_r(\tau, u) = \sum_i(\lambda_i - 1) A_i(\tau, u_i)$. QPlex places restrictions $A_i(\tau, u_i) = Q_i(\tau, u_i) - max_{u_i}Q_i(\tau, u_i)$ among the main and the residual function, and the restrictions imply that $A_i \leq 0$. In contrast, ResQ directly requires $Q_r(\tau, u) \leq 0$. Further, ResQ can use more expressive mixers (such as QMIX) to model $Q_{tot}$ than QPLEX (it uses VDN for $Q_{tot}$).

In summary, ResQ places fewer restrictions on the relationships between the main and the residual function than QPLEX. And ResQ can use a more expressive neural network to model the main function than QPLEX. Thus, ResQ can model Q value function in a better and more flexible way than QPLEX.

CW/OW QMIX can be viewed as variants of ResQ . CW/OW QMIX learns $Q_{wqmix} = w_{tot}(\tau, u) Q_{tot}(\tau, u) + (1-w_{tot}(\tau, u)) Q_r(\tau, u)$ to approximate the true value function, where $w_{tot}(\tau, \bar{u}) = 1$ and $w_{tot}(\tau, u) = 0 \quad \forall u \neq \bar{u}$. They assign high learning priorities to $Q_{tot}$, which puts the learning of sub-optimal state-action pairs $Q_r$ and $Q_{wqmix}$ in trouble.

**Limitation**
Knowing the optimal action over Q is computationally intractable, we make approximations to derive a practical algorithm. It is difficult for ResQ to find the optimal actions for a scenario requiring highly-coordinated agent exploration. The approximated mask function will fail if the approximated optimal action differs significantly from the optimal action. In ResQ, we assume that the argmax operator of $Q_{tot}$ can lead to correct optimal actions. However, this assumption does not always hold. For a discrete-action environment with a tabular Q value, we think that all state-action values can be factorized using ResQ if the masking function can find the correct optimal action. As we use neural networks to represent states and actions, the approximation error of neural networks can interact with the error of the approximated argmax operator. This may make ResQ fail. We think that combining efficient-exploration approaches (e.g., MAVEN) with ResQ can make the mask function more robust than using ResQ alone.

---

> ### Comment · Reviewer_dP9n · 2022-08-03
> **Response to the authors**
>
> According to the authors, ResQ can model suboptimal state-action pairs better than Qtran resulting in better performance. In general, I disagree with this claim. The $Q_{jt}$ in ResQ can model the true Q-value well or the $Q_r$ can well compensate the difference between the true Q-value and $Q_{tot}$. It’s true. But the $Q_{jt}$ of Qtran can also model the true Q-value well since it has no representation limitations.
>
> What really matters are the properties of $[Q_i]$s which are directly related to the action selection. $Q_i$, on the other hand, cannot capture both optimal and sub-optimal state-action pairs simultaneously. For example, in tabular setting, to represent the Q-value requires $|S||A|^n$ parameters, whereas $[Q_i]$s only contain $n|S||A|$ parameters. As a result, $Q_i$ must sacrifice some suboptimal actions in order to fit the optimal actions. In this way, the sum of $Q_i$ (i.e. $Q_{jt}'$) in Qtran will overestimate some sub-optimal actions, not the overestimation of $Q_{jt}$ itself. Similarly, the sum (or mix) of $Q_i$ in ResQ, i.e. the $Q_{tot}$, will also overestimate some sub-optimal actions. The difference is that ResQ uses $Q_r$ to compensate the difference between the true Q-value and $Q_{tot}$ while Qtran ignores this part. The authors mistakenly compare the $Q_{jt}$ in ResQ with the $Q_{jt}'$ in Qtran and conclude that ResQ matches the sub-optimal actions better.
>
> In summary, both Qtran’s and ResQ’s $Q_{jt}$ can model the true Q-value well, and Qtran’s $Q_{jt}'$ and ResQ’s $Q_{tot}$ will overestimate some sub-optimal actions. The theoretical property of $Q_i$ is essential for maintaining optimal consistency. However, ResQ does not have a better guarantee than Qtran. Similar for Qplex, whether “ResQ can use more expressive mixers” or “places fewer restrictions” on the residual function does not really matter, because the theoretical property of $Q_i$ is the same. With the same guarantee of optimality, one could even argue that a more restricted architecture has smaller parameter space and is thus easier to learn.

---

> > ### Author Response · Authors · 2022-08-03
> > **Response to Reviewer 4 (dP9n)**
> >
> > Thanks for the reviewer's quick response and insightful comments. As it is written in the reviewer's comment *"ResQ uses $Q_r$ to compensate the difference between the true Q-value and $Q_{tot}$ while Qtran ignores this part."*, we want to thank the reviewer for his agreement that the residual function can help ResQ model true Q better than QTran. I think we have addressed the question raised by the reviewer's initial questions *"what is the benefit of learning the residual function"*, *"better theoretical guarantees than Qtran"*, and *"why ResQ perform better"*.
> >
> > We would like to thank the reviewer for not disagreeing that ResQ is a generalization of QTran, QPlex, CW/OW QMIX. This is a contribution of ResQ, as different designs of the main function, mask functions, and residual functions could be developed by using the idea of residual functions. We believe that using the concept of residual functions will be beneficial to MARL, we expect that there could be more value factorization methods using residual functions developed in the future.
> >
> > We would like to thank the reviewer for not disagreeing that using residual functions can satisfy the IGM and the DIGM theorems without representation limitations. QTran, QPlex, CW/OW QMIX deal with the IGM theorem only, and DMIX satisfies the DIGM theorem with representation limitations.
> >
> > There may be some misunderstandings regarding QTran and ResQ. QTran learns a value function $Q_{tran}$  to approximate the true value function $Q$. $Q_{tran}$ does over-estimate values for sub-optimal actions, but QTran does not over-estimate the true value function $Q$ (it is defined as $Q_{jt}$ in QTran). In ResQ, we learn $Q_{ResQ}$ (denoted as $Q_{jt}$ in ResQ) to approximate the true value function $Q$. As it is shown in Theorem 2, $Q_{ResQ}$ does not over-estimate value for sub-optimal actions.
> >
> > To discuss the relationship with QTran, We have copied Theorem 1 of QTran as follows (with adaptation to suit openreview input format).
> >
> > **Theorem 1 of QTran**
> > A factorizable joint action-value function $Q_{jt}(\tau,u)$ is factorized by $[Q_i(\tau_i , u_i)]$, if
> > $\sum_{i=1}^N Q_i(\tau_i, u_i) - Q_{jt}(\boldsymbol{\tau}, \boldsymbol{u}) + V_{jt}(\boldsymbol{\tau}) = 0 \quad \boldsymbol{u}=\bar{\boldsymbol{u}}, \quad (4a)$ and $\sum_{i=1}^N Q_i(\tau_i, u_i) - Q_{jt}(\boldsymbol{\tau}, \boldsymbol{u}) + V_{jt}(\boldsymbol{\tau}) \geq 0 \quad \boldsymbol{u} \neq \bar{\boldsymbol{u}}, \quad (4b)$
> > where $V_{jt} = max_u Q_{jt}(\boldsymbol{\tau}, \boldsymbol{u}) - \sum_{i=1}^N Q_i(\tau_i, \bar{u}_i).$
> >
> > In this theorem, $Q_{jt}(\boldsymbol{\tau}, \boldsymbol{u})$ is the true value function, QTran does not over-estimate it. The approximated value function $Q_{tran}$  (following the definition of QTran++ which is written by the same authors) is defined as $Q_{tran}(\boldsymbol{\tau}, \boldsymbol{u})= \sum_{i=1}^N Q_i(\tau_i, u_i) + V_{jt}(\boldsymbol{\tau})$. From (4b) of this theorem,  $\sum_{i=1}^N Q_i(\tau_i, u_i) - Q_{jt}(\boldsymbol{\tau}, \boldsymbol{u}) + V_{jt}(\boldsymbol{\tau}) \geq 0 \quad u \neq \bar{u}$. That is, $Q_{tran}(\boldsymbol{\tau}, \boldsymbol{u}) \geq Q_{jt}(\boldsymbol{\tau}, \boldsymbol{u}) \quad \boldsymbol{u} \neq \bar{\boldsymbol{u}}$. $Q_{tran}$ is the function learned by QTran to approximate $Q_{jt}$. Clearly, $Q_{tran}(\boldsymbol{\tau}, \boldsymbol{u})$ can over-estimate the true value function $Q_{jt}(\boldsymbol{\tau}, \boldsymbol{u})$ for sub-optimal actions. In QTran, $Q_{jt}' = \sum_i^N Q_i(\tau_i, u_i)$ is called ``transformed joint-action value function''. It is analogous to $Q_{tot}$ in ResQ. We will improve the writing of this work to avoid confusion.
> >
> > ResQ, QTran, QPlex, and CW/OW QMIX are value factorization methods within the centralized training with decentralized execution regime. They use $[Q_i]$ for execution, and they suffer from the same theoretical limitation of $Q_i$. However, these $[Q_i]$ are trained with the help of value factorization methods (mixer functions). Better mixer functions can lead to better $[Q_i]$ for execution.
> >
> > We agree with the reviewer that the upper bound for ResQ, QTran, QPlex, and CW/OW QMIX is to find the optimal policies which satisfy the IGM theorem. As we use neural network approximators for functions in ResQ, we will soften the claim of ResQ as it could find policies that approximate the IGM and the DIGM theorems and the experimental results are promising.

---

> > > ### Author Response · Authors · 2022-08-05
> > > **Response to comparison with QTran and QPlex**
> > >
> > > To avoid miss-understanding, let us explain the over-estimation issue of QTran further. For a non-monotonic pay-off value function $Q$ presented in the QTran paper listed as follows.
> > >
> > >   8  | -12 | -12
> > >
> > >   -12|  0  |  0
> > >
> > >   -12|  0  |  0
> > >
> > > First, QTran learn a value function $Q_{appro}$ to approximate the true value function $Q$. And then QTran learns a function $Q_{tran}$ to approximate $Q_{appro}$. $Q_{tran}=\sum_{i=1}^n Q_i(\tau_i, u_i) +V(\tau)$ (We called it as $Q_{tran}$ following the definition of the QTran++ paper by the same author, it was called $Q_{jt}$ of QTran)
> > >
> > > We have trained QTran on $Q$ for 50,000 steps. The $Q_{appro}$ learned by QTran to approximate $Q$ is listed as follows.
> > >
> > >   8.00   | -12.01| -12.01
> > >
> > >   -12.01 | -0.00 | -0.00
> > >
> > >   -12.01 | -0.00 | -0.00
> > >
> > > $Q_{appro}$ approximates $Q$ well. This matches the result shown in Table 1 c of the Qtran paper.  However, $Q_{tran}$ **does not** approximate $Q_{appro}$ and the true value function $Q$ well. The $Q_{tran}$ learned by QTran is listed as follows.
> > >
> > >   8.00 | 4.44 | 4.41
> > >
> > >   5.38 | 1.83 | 1.80
> > >
> > >   5.32 | 1.76 | 1.73
> > >
> > > And the $Q_i$ learns by QTran is $Q_1 = [3.31,  0.69, 0.62]$, $Q_2 = [4.49, 0.93, 0.90]$, and the $V(\tau)$ is 0.20.
> > >
> > > As we can observe from the matrix, $Q_{tran}$ **does over-estimate** the true value function $Q$ for all sub-optimal actions. $Q_{tran}$ can be written as $Q_{tran} = Q_{jt}' + V$, where $Q_{jt}' = \sum_{i=1}^n Q_i$. Table 1 (b) of the QTran paper confirms that $Q_{jt}'$ does over-estimate values as well.
> > >
> > > The value function $Q_{appro}$ learned by ResQ  to approximated the true value function $Q$ is listed as follows.
> > >
> > >   8.01   | -11.92 | -11.92
> > >
> > >   -11.91 | 0.01   | 0.01
> > >
> > >   -11.91 | 0.01   | 0.01
> > >
> > > $Q_{appro}$ is very close to the true value function.
> > >
> > > And the $Q_{jt}$ (called $Q_{ResQ}$ as well) learned by ResQ to approximate $Q_{appro}$ is list as follows.
> > >
> > >   8.00   | -12.00 | -12.00
> > >
> > >   -12.00 | -0.00  | -0.00
> > >
> > >   -12.00 | -0.00  | -0.00
> > >
> > > $Q_{ResQ}$ approximates the $Q_{appro}$ and $Q$ well, and it does not over-estimate value for sub-optimal actions.
> > > The utility function learns by ResQ is $Q_1 = [1.43, -2.14, -2.11]$ and $Q_2=[1.42, -2.14, -2.11]$.
> > >
> > > For QPlex, it does not learn a surrogate approximation function $Q_{appro}$ to approximate the true value $Q$, and it learns $Q_{plex}$ to approximate the true value $Q$ directly. $Q_{plex}$ is listed as follows.
> > >
> > >   26.57 | 6.33  | 6.73
> > >
> > >   8.25  | 19.05 | 18.95
> > >
> > >   8.28  | -0.00 | 19.09
> > >
> > > $Q_{plex}$ is quite different from $Q$. And it is not stable, $Q_{plex}$ can change quickly for each 1,000 training steps.
> > > $Q_{plex} = V + Adv$, where $V = max_uQ$ and $Adv = Q - V$.  During learning $Q_{plex}(u, v)$ try to match $r + \gamma max_u(Q_{plex}(u, v)) = r +\gamma max_u(max_uQ + Adv)$. As there are two max operators during the update of QPlex, we think that this cause its learning instability.

---

### Meta-Review · Area_Chair_cRT6 · 2022-08-27

**Recommendation:** Accept
**Confidence:** Certain

**Metareview:**

The paper is for the most part well written and contains both theoretical analyses and a comprehensive empirical study. One of the main initial concerns brought up by various reviewers is that the relation between the proposed method, resQ, and the closely related existing methods Qtran and Qplex is not 100% clear. The authors addressed this point extensively in the rebuttal. However, the theoretical advantages of resQ over Qtran remains unclear even after the rebuttal phase for one of the reviewers (despite promising empirical results)
Overall, I believe the paper's strengths make up for this potential weakness and recommend acceptance. I do want to recommend that the authors take a careful look at reviewer  dP9n's comments and clarify any points of confusion in the final version of this paper.



**Award:**

No

---

### Decision · Program_Chairs · 2022-09-14

Accept